# Coordinated multi-level adaptations across neocortical areas during task learning

Shuting Han [1,2,3] ✉ & Fritjof Helmchen [1,2,3] ✉

The coordinated changes of neural activity during learning, from single neurons to populations of neurons and their interactions across brain areas, remain poorly understood. To reveal specific learning-related changes, we applied multi-area two-photon calcium imaging in mouse neocortex during training of a sensory discrimination task. We uncovered coordinated adaptations in primary somatosensory area S1 and the anterior (A) and rostrolateral (RL) areas of posterior parietal cortex (PPC). At the single-neuron level, task-learning was marked by increased number and stabilized responses of task neurons. At the population level, responses exhibited decreased dimensionality and reduced trial-to-trial variability, paralleled by enhanced encoding of task information. The PPC areas became gradually engaged, opening additional within-area subspaces and inter-area subspaces with S1. Task encoding subspaces gradually aligned with these interaction subspaces. Behavioral errors correlated with decreased encoding accuracy and misaligned subspaces. Thus, multi-level adaptations within and across cortical areas contribute to learning-related refinement of sensory processing and decision-making.

Learning a new task requires accurate sensory processing and transforming relevant sensory information into appropriate actions. This process often involves not only reconfigurations of local neuronal populations but also coordinated changes across brain areas. Many studies have characterized learning-related changes at single-neuron and population coding levels within local populations[1–7]. These changes include increased stimulus selectivity[1,6], recruitment of new neurons[3,6], stabilized responses of individual neurons[2,3,7], and increased task information encoded by the populations[1–6]. Recently, an increasing number of studies focused on understanding how population activity dynamics, which often reflects task-related information, changes with learning[8–12]. For example, learning a new task is accompanied by stabilized population dynamics and optimized activity subspaces of intrinsic dynamics to encode task-relevant variables[10,13–15]. The local population dynamics can also constrain the learning process: learning something new within the intrinsic subspace of neural dynamics is easier than outside of the subspace[16], and the internal state of the animal, e.g., the level of task engagement, can shape learning through changing these subspaces[17]. Understanding such population-level changes is important for unveiling the neural computations underlying the optimization of sensory processing during learning, supported by changes on the single-neuron level.

Beyond the changes in local populations, task-learning is often accompanied by coordinated changes across brain areas. Learning to perform a specific task typically involves a distinct set of brain areas, with specific higher areas becoming increasingly engaged to selectively process task-relevant information sent by primary sensory areas[6,18,19]. However, we still poorly understand how information processing is transformed in a coordinated way across multiple areas during learning, from primary sensory areas that receive sensory inputs, to higher areas that further process sensory information, make decisions, and generate actions. Despite the rising interest in understanding cross-areal processing from the perspective of behavior-relevant subspaces as well as interaction subspaces[20–22], it remains

[1]Brain Research Institute, University of Zurich, Zurich, Switzerland. [2]Neuroscience Center Zurich (ZNZ), University of Zurich, Zurich, Switzerland. [3]University Research Priority Program (URPP), Adaptive Brain Circuits in Development and Learning, University of Zurich, Zurich, Switzerland. ✉e-mail: han@hifo.uzh.ch; helmchen@hifo.uzh.ch

unclear whether within-area and inter-area interaction subspaces are reconfigured during learning to facilitate task information processing, or whether these intrinsic subspaces rather impose constraints on learning. One major challenge to answer these questions is to record from enough neurons simultaneously across multiple areas. Here, we addressed this challenge by employing a custom multi-area two-photon microscope[23,24] to measure the learning-related changes in population dynamics, within and between areas along the cortical hierarchy, and to analyze the changes in their interaction subspaces.

A key area in processing and routing sensory information is the posterior parietal cortex (PPC). The PPC has dense connections with primary sensory regions, including the visual (V1), somatosensory (S1), and auditory (A1) cortices, as well as with frontal areas and the thalamus[25]. PPC supports a wide range of functions, such as multisensory integration, evidence accumulation, decision-making, and working memory[26–29]. Although PPC overall receives inputs from all primary sensory regions, recent works have supported the existence of functional subregions of PPC in the mouse brain: the rostrolateral region (PPC-RL) receives more inputs from V1 and S1, and is involved in visual and tactile processing, and the anterior region (PPC-A) receives more inputs from V1 and A1, and is involved in visual and auditory processing[23,26]. In particular, PPC emerges as a key area for routing sensory information during learning of a texture discrimination task[26,28,30], and is dynamically reorganized during the learning process[8]. These features make PPC an attractive example area to study the principles of learning mechanisms within and across areas.

Here we characterize learning-related changes in the local populations of whisker-related S1 barrel cortex, PPC-RL, and PPC-A. Specifically, we trained mice to perform a two-alternative auditory-cued texture discrimination task, while simultaneously imaging population activity in S1 and PPC throughout learning. To provide an overview of learning-related changes from this rich dataset, we systematically examined four types of population subspaces: the variance subspace, the encoding subspace, the within-area interaction subspace, and the inter-area interaction subspace. We find that task-learning is accompanied by systematic and coordinated changes within and across these areas, both on the single-neuron level and on the population level. The PPC areas were gradually engaged during the learning process, expanding their within-area and inter-area interaction subspaces with S1. Task representation was amended through an improved alignment of encoding subspaces with these interaction subspaces. These learning-related changes were degraded during incorrect trials. These results suggest that refinement of sensory and choice processing during learning is achieved through coordinated adaptations across neocortical areas on multiple levels, from the enhanced responses of task neurons, to improved alignment of task-relevant information, to the optimized intrinsic dynamic subspaces within and across areas.

## Results

### Behavioral task

We trained 15 mice (age 2–4 months, both sexes) expressing GCaMP6f in L2/3 to perform an auditory-cued two-alternative texture discrimination task while monitoring neuronal population activity in S1 and PPC[23] (Fig. 1a and Methods). The task consisted of four sequential phases (Fig. 1b and Methods): each trial started with an auditory tone (tone window), followed by presentation of sandpaper texture to the whisker pad (texture window). A low tone was always paired with rough sandpaper and a high tone with smooth sandpaper. Then, the mouse chose between two lick ports (choice window) to obtain a sugar water reward (reward window). The reward window was triggered as soon as mice had made their choice. Each day, mice performed between 100 and 500 trials during training. To account for the different trial numbers each day, we split each training day into sessions of 80–120 trials (Methods). To study learning-related changes, we defined three learning phases according to the behavioral

performance: naive (performance<55%), learning (performance 55%–75%), and expert (performance ≥75%) phase (Fig. 1c). During the expert phase, a subset of the sessions included a small fraction (10–30%) of tone-texture mismatch trials, in which the tone was paired with the non-matching texture, and reward was given according to texture. These data revealed interesting results about predictive processing and have been previously published[23]. Due to the low percentage of mismatch trials and because we did not observe re-learning due to these mismatch trials, we included this expert dataset for analysis, but excluded all the mismatch trials, as well as the sessions with low behavioral performance (<70% correct rate).

Task-learning was accompanied by various behavioral changes. Compared to naive sessions, mice showed reduced licking in the tone and early texture windows in expert sessions (Fig. 1d, e), as well as reduced early responses (Fig. 1f). These changes were present in both correct and incorrect trials (Fig. 1d–i), suggesting a systematic change of the behavioral strategy of mice. We also monitored the pupil and movement of mice with a behavior camera (Fig. 1a). The pupil diameter during the reward window was larger in expert compared to naive sessions (Fig. 1j), and face movements were reduced during the tone window, consistent with the reduced licking behavior (Fig. 1k). Removing the data points during early licks (licks before choice window) led to a comparable level of face movement during the tone window across learning phases (Supplementary Fig. 1). These observations suggest a refinement of the task-related behavior of the mice over the learning process.

### Recruitment of task neurons in S1 and PPC during learning

To study learning-related changes across the neocortical areas, we used a custom-built two-area two-photon microscope to simultaneously record from neuronal populations in S1 barrel cortex and the two PPC subregions[23,24]. In our experiments, we simultaneously imaged somatic calcium signals from S1 and PPC-RL (total 13 mice) or from S1 and PPC-A (total 9 mice), from 2–3 depths (100–300 μm) in L2/3, covering 50–600 neurons in each area (Fig. 2a and Methods). The exact locations of S1, PPC-RL and PPC-A were identified by sensory mapping and retinotopic mapping (Methods). We extracted ΔF/F traces as well as the deconvolved spike rates of individual neurons using Suite2p[31], and all following analysis concerning neuronal activity was performed on the z-scored spike rates. The z-scoring procedure, performed independently for each session, in principle removed the effect of changes in population firing rates across learning for all the following analysis. Over the course of training, we could follow roughly the same field of views (FOVs) with similar imaging quality (Fig. 2b, c). However, due to restrictions by the dense labeling in the GCaMP6f transgenic mouse lines, the large FOVs across multiple depths, and the long training process, we did not seek to match and track individual neurons across days for each mouse.

We first examined whether PPC-RL and PPC-A were required for performing the task using optogenetic inhibition. We trained 8 mice expressing channelrhodopsin-2 (ChR2) in GABAergic neurons (VGAT-ChR2-EYFP transgenic line) to perform the task. Once they had reached the expert phase, we tested their performance when PPC areas were bilaterally photoinhibited through activation of GABAergic neurons (Supplementary Fig. 2a). Inhibition of either PPC-RL or PPC-A during the texture window reduced behavioral performance (Supplementary Fig. 2b), suggesting that both areas are required for optimally performing the task. Additionally, we also inhibited PPC areas in tone-only and texture-only conditions, where only one stimulus modality was presented to the mice. Inhibition of either PPC-RL or PPC-A reduced performance in the tone-only condition where no texture was presented (Supplementary Fig. 2c), suggesting that both areas were involved in transforming sensory association to decision[23]. Inhibiting PPC-RL, but not PPC-A, also impaired texture-only performance (Supplementary Fig. 2d), suggesting that PPC-RL was more involved in

processing pure texture information. These results indicate that both PPC areas are required for transforming sensory information adequately into correct actions in this task.

Task-learning can be accompanied by activity changes in neurons within local populations. While some studies have reported an increased number and enhanced responses of neurons with task-related activity (defined as 'task neurons') during learning in the neocortex, others have found the opposite[1,6,7,18]. Therefore, we first characterized the changes in task neurons in S1 and PPC during the learning process. We identified two types of task neurons: task-responsive neurons were defined as neurons that showed significantly higher activity in specific task windows compared to randomly sampled activity from a matching number of frames outside of the window; task-discriminative neurons were defined as a subset of responsive neurons that showed a significant difference in activity with respect to the relevant task variable in the specific window (tone 1 vs. tone 2; texture 1 vs. texture 2; lick left vs. lick right; reward left vs. reward

right). These neurons were identified separately for each task window and the corresponding task variables. Since early licks during the texture window were not punished, we removed all neural activity frames from the first early lick until the choice window onset in all following analysis, to avoid the confounding influence of licking-related activity. As the first lick during the choice window triggers outcome, we further aligned each trial to this lick time and defined the choice window as the 0.5-s period before the first lick in all following analysis. Overall, S1 displayed a higher percentage of texture-responsive neurons compared to other task variables, and this percentage further increased during learning, highlighting the role of S1 in texture processing (Fig. 2d). Whereas PPC-RL showed an increased percentage of texture-responsive neurons, PPC-A developed a higher percentage of tone-, texture- and choice-responsive neurons, indicating its role as a higher-order area in this task. Task-discriminative neurons in these areas also increased during learning, with PPC-A overall containing the highest percentage of task-discriminative

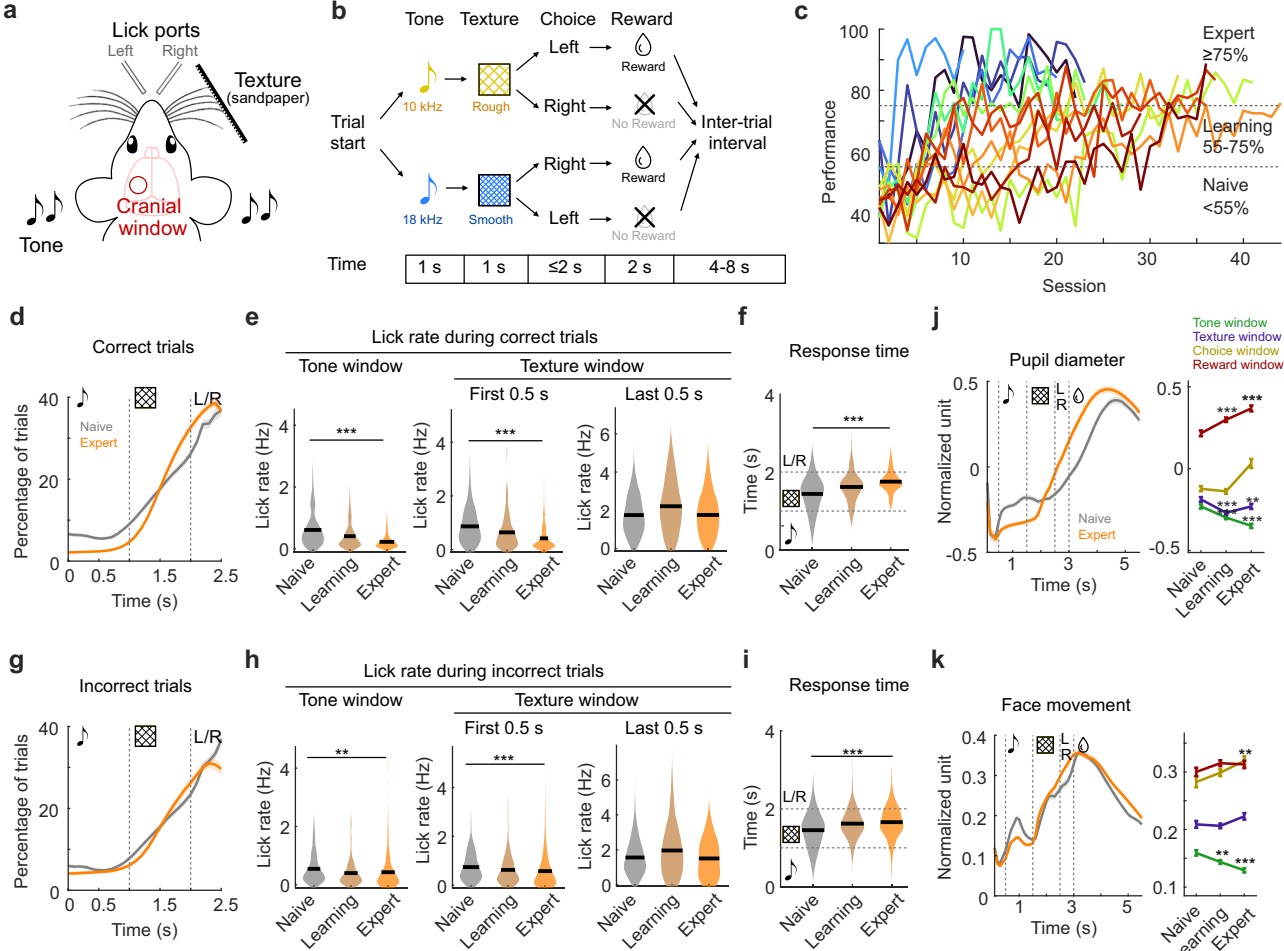

Fig. 1 | **Learning behavior of mice in a two-alternative tone-texture discrimination task. a** Schematic of Experimental setup (adapted from reference 23). **b** Task design. Each trial consisted of a tone followed by a paired texture presented to the whisker pad of the mice. Then, mice chose between two lick ports to receive a sugar water drop as reward. **c** Learning curves of all mice (n = 16), subdivided into sessions of 80–120 trials. Naive is defined as performance <55%, learning as 55–75% and expert as ≥75%. Line colors indicate mice. **d** Lick pattern in correct trials, represented as percentage of trials with a lick at each time point, in naïve (gray line) and expert (orange line) mice. **e** Quantification of lick rates during tone window (left), early texture window (middle), and late texture window (right), in correct trials (black line represents mean). **f** Response time in correct trials. **g** Lick pattern in incorrect trials, represented as percentage of trials with lick at each time

point, in naive and expert mice. **h** Quantification of lick rates during tone window (left), early texture window (middle), and late texture window (right), in incorrect trials. **i** Response time in incorrect trials. **j** Pupil diameter (z-scored) over trial time (left) and quantification over learning (right). Gray and orange in the left panel indicate naive and expert conditions; colors in the right panel indicate task windows. **k** Face movement (z-scored) over trial time (left) and quantification over learning (right). Colors as in (**j**). (15 mice, **d–i**: n = 138 [naive], 192 [learning], 171 [expert] sessions; **j**, **k**: n = 134, 191, 166 sessions due to missing/broken behavior videos; two-sided Wilcoxon rank-sum test; *p < 0.05, **p < 0.01, ***p < 0.001; p-values were corrected for multiple comparisons using FDR method in **j**, **k**; mean ± SEM). Source data are provided as a Source Data file.

neurons for tone, choice and reward (Fig. 2e). In particular, the percentage of task-discriminative neurons that were jointly responsive to adjacent task windows also increased (Supplementary Fig. 3), potentially benefiting the transition between task phases and allowing for

more robust population dynamics during the trials. We also investigated the temporal response profile of these task neurons over the course of learning by comparing the average firing rate of the task-discriminative neurons in naive and expert sessions. As the percentage

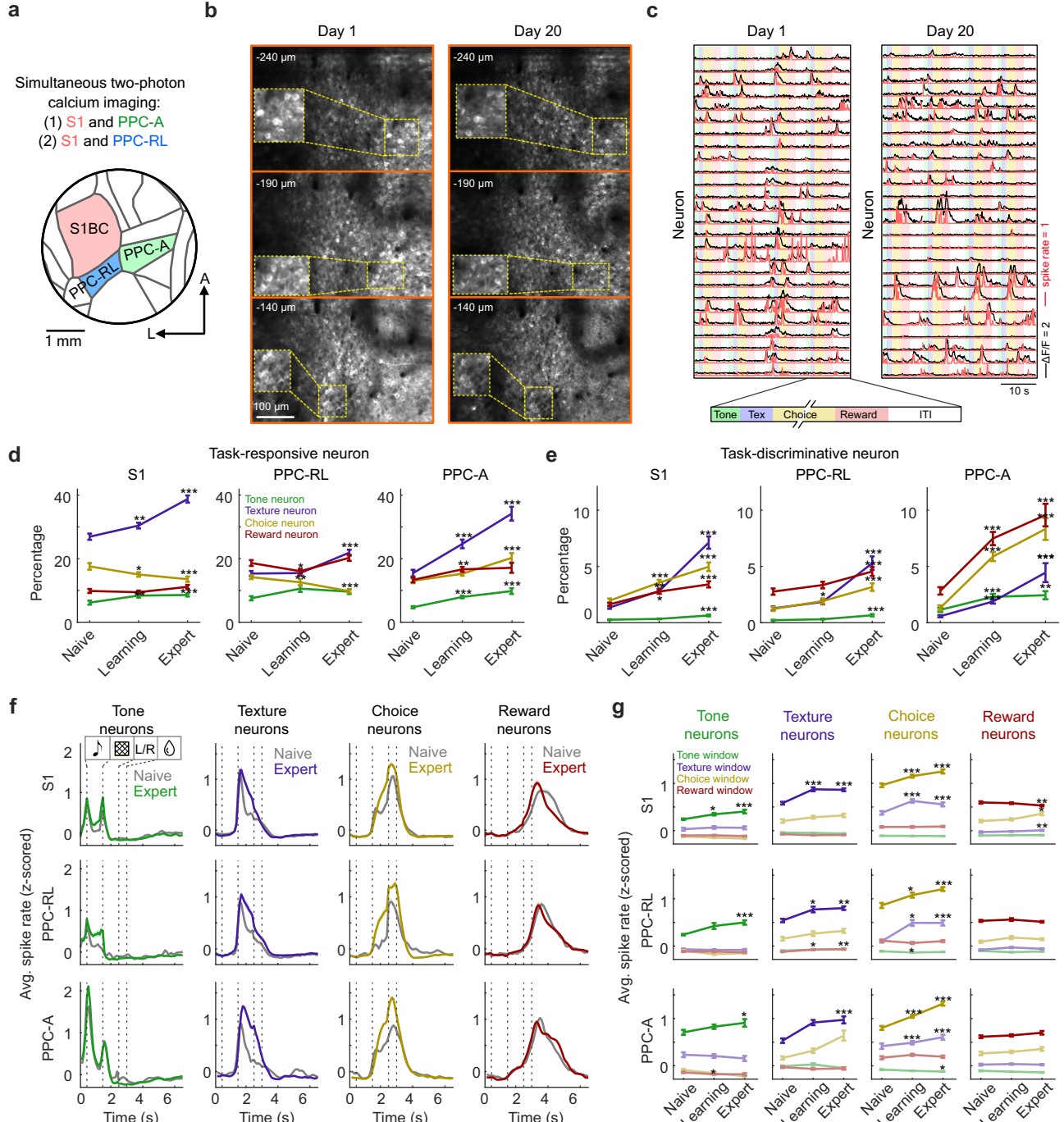

**Fig. 2 | Task neurons in S1 and PPC areas. a** Diagram of the cranial window and S1/PPC locations. Simultaneous two-photon calcium imaging was performed in S1 and PPC areas in GCaMP6f transgenic mice. **b** Example FOVs at 3 imaging depths from day 1 and day 20 of the same mouse, same area (PPC-RL). **c** Example neuronal activity from day 1 and day 20, from the same mouse and the same area as shown in (**b**). Colored stripes in the background represent task windows. Black lines represent ΔF/F traces, red lines represent deconvolved spike rates in arbitrary units. **d** Percentage of task-responsive neurons in S1, PPC-RL, and PPC-A. Colors represent the task window to which the neurons are responsive. **e** Percentage of task-discriminative neurons in S1, PPC-RL, and PPC-A. Colors represent the task window to which the neurons are responsive. **f** Session-average spike rates of task-

discriminative neurons over trial time. Spike rates were z-scored within each session. Gray lines represent the naive condition, colored lines represent the expert condition. Dashed vertical lines indicate task windows. **g** Quantification of average spike rates of task-discriminative neurons, in different task windows. Colors represent the quantified task window. (S1: 13 mice, 130, 190, 153 sessions [naive, learning, expert]; PPC-RL: 13 mice, 70, 77, 104 sessions; PPC-A: 9 mice, 60, 111, 45 sessions; *$p < 0.05$, **$p < 0.01$, ***$p < 0.001$; Statistical tests in (**d**, **e**, **g**) were performed using two-sided Wilcoxon rank-sum test between Naive and Learning sessions, and between Naive and Expert sessions, using pooled session data; $p$-values were corrected for multiple comparisons using FDR method; mean ± SEM). Source data are provided as a Source Data file.

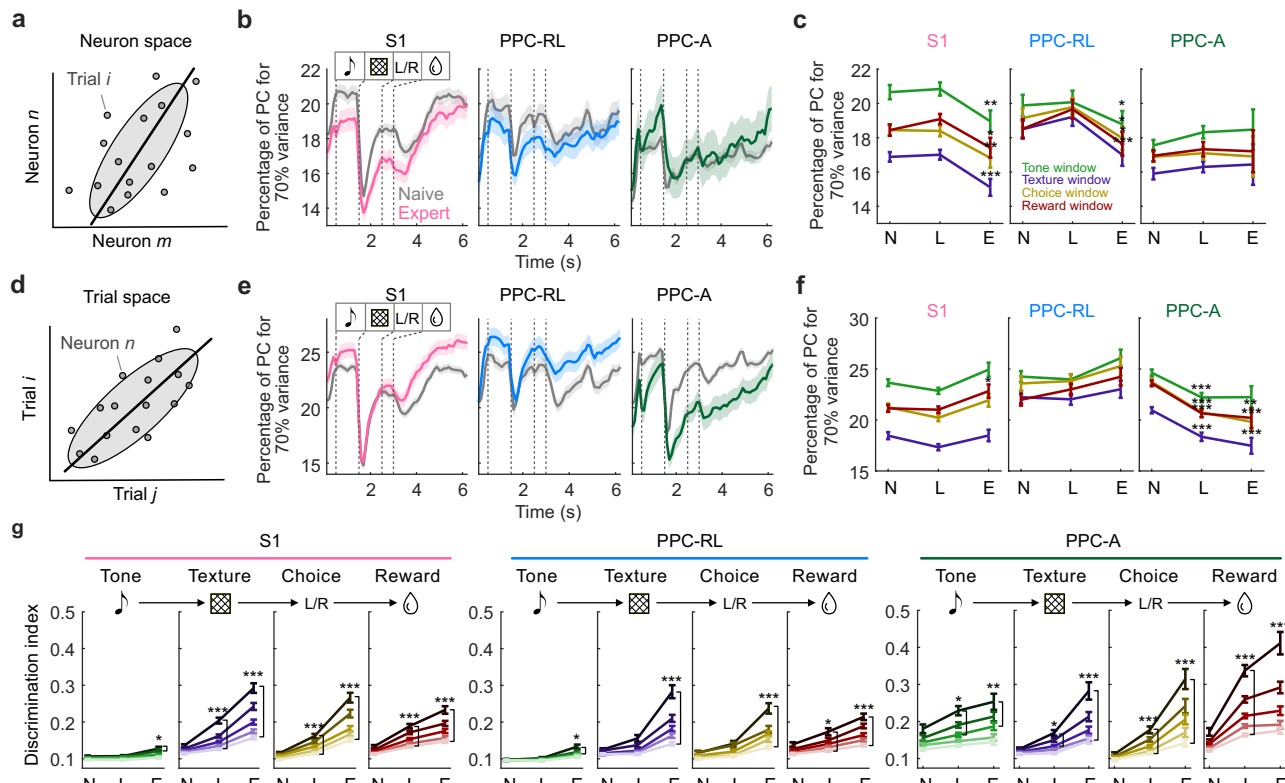

**Fig. 3 | Decreased neuronal dimensionality and trial variability during learning.**
**a** Diagram of neuron variance space. PCA was performed with neurons as variables.
**b** Percentage of principle components (PCs) that explains 70% of variance in neuron subspace, over the trial structure. Vertical dashed lines indicate task windows. Colors represent learning phases. **c** Quantification of the percentage of PCs in neuron space during each task window. Colors represent task window. **d** Diagram of trial variance space. PCA was performed with trials as variables. **e** Percentage of PCs that explains 70% of variance in trial space, during the trial time. Vertical dashed lines indicate task windows. Colors represent learning phases.
**f** Quantification of the percentage of PCs in trial space, during each task window.

Colors represent task window. **g** Discrimination index of the top 5 PCs during each task window, regarding the corresponding task variables. Line color saturation represents the PC number. Naïve, N; learning; L, expert, E. (S1: 13 mice, 130 [N], 183 [L], 138 [E] sessions; PPC-RL: 13 mice, 70, 74, 98 sessions; PPC-A: 9 mice, 60, 109, 40 sessions; sessions with less than 30 neurons on average across the resampling procedure were excluded; *$p < 0.05$, **$p < 0.01$, ***$p < 0.001$; two-sided Wilcoxon rank-sum test against naive condition; $p$-values were corrected for multiple comparisons using FDR method; mean ± SEM. Source data are provided as a Source Data file.

of task neurons increased during learning, their average response profile also broadened (Fig. 2f), suggesting a tiling of task neuron activity during the trials, contributing to longer-lasting and more stable population dynamics across S1 and PPC areas. Additionally, the average response strength of texture and choice neurons during the texture and choice window also increased (Fig. 2g). Thus, task-learning was accompanied by an overall recruitment of task neurons as well as extended and enhanced neuronal responses.

### Reduced population response dimensionality and trial variability

Performing a task involves transforming single-neuron activities into a unified sensory representation to guide decisions. During the task, neuronal population activity often resides in a low-dimensional subspace, allowing for robust information representation and reliable behavioral output[12,32–35]. To further understand the changes in the population activity structure regarding variability across neurons and trials, we analyzed the dimensionality of the neuron space and the trial space. To account for the reduced number of recorded neurons in the expert phase (due to the decay in imaging quality over longitudinal imaging), we resampled the neuronal populations in naive and learning sessions to match one of the expert sessions and repeated this procedure 50 times. For the dimensionality of the neuron space, we performed principal component analysis (PCA) on the population activity, with neurons as variables and trials as observations at each time point

of the trial, for each session separately (Fig. 3a). This analysis identifies coordinated changes across neurons and the dominant patterns in the activity. Compared to naive sessions, the population dimensionality in expert sessions consistently decreased in S1 and PPC-RL areas in all task windows but remained unchanged in PPC-A (Fig. 3b, c). This result suggests that in expert sessions, neurons in S1 and PPC-RL developed correlated firing patterns in each task window, while the activity patterns in PPC-A remained diverse. We further analyzed the dimensionality in the trial space by performing PCA on the population activity with trials as variables and neurons as observations (Fig. 3d). This analysis directly measures the neuronal reliability across trials. While the dimensionality remained unchanged in S1 and PPC-RL during the learning process, it decreased in PPC-A (Fig. 3e, f), indicating decreased trial-to-trial response variability.

We wondered if the changes in neuronal dimensionality were beneficial for carrying task information. To test this, we performed a decoding analysis on the top 5 principal components (PCs) of the neuron space during each task window, using the corresponding task variables (e.g., decoding texture identity during the texture window). All top 5 PCs across S1 and PPC areas showed an increased discrimination index of task variables over the course of learning (Fig. 3g), suggesting improved task information encoding across multiple dimensions. We conclude that task-learning was accompanied by reduced trial-to-trial variability and enhanced task encoding in S1 and PPC populations.

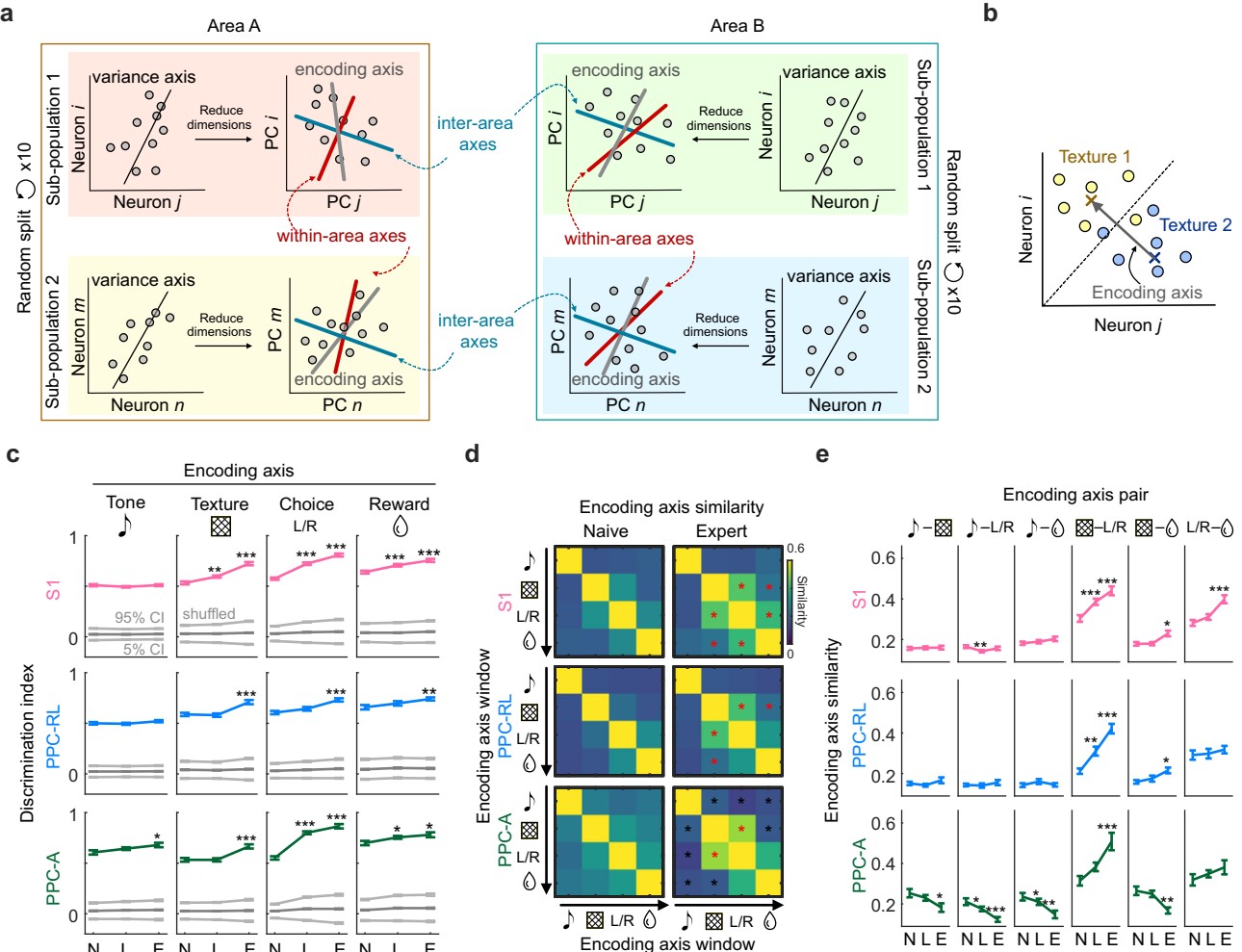

**Fig. 4 | Improved task representation in S1 and PPC encoding subspaces during learning. a** Diagram of generating encoding axis, within-area interaction axis, and inter-area interaction axes in the same reduced 30-dimensional neural space for each subpopulation. **b** Example encoding axis scheme for the texture axis. Colored dots represent neural activity in texture 1 and texture 2 trials, during the texture window. Crosses represent the means of responses for the two texture types. **c** Discrimination index of population activity projection on the encoding axes, for each task variable, in S1, PPC-RL, and PPC-A. Gray lines represent confidence intervals (CI) of shuffled data. **d** Color-coded pairwise similarity of encoding axes

for all task variable combinations. Symbols on the x- and y-axis indicate task windows. Red and black asterisks indicate that the data from the expert condition is significantly higher or lower than the naive condition. **e** Quantification of pairwise encoding axes similarity. (S1: 13 mice, 122, 171, 119 sessions [naïve, N; learning, L; expert, E]; PPC-RL: 13 mice, 66, 72, 86 sessions; PPC-A: 9 mice, 56, 99, 33 sessions; (c and e): *$p < 0.05$, **$p < 0.01$, ***$p < 0.001$; (d): *$p < 0.05$; two-sided Wilcoxon rank-sum test against naive condition; $p$-values were corrected for multiple comparisons using FDR method; mean ± SEM). Source data are provided as a Source Data file.

## Improved task encoding subspaces in S1 and PPC populations during learning

In addition to the global changes in the variance subspace during task-learning, the representation and processing of task information can also be improved through coordinated changes across areas. To examine this aspect, we systematically investigated three additional types of population subspaces during the task: the encoding subspace, the within-area interaction subspace, and the inter-area interaction subspace (Fig. 4a). The encoding subspace captures the optimal encoding direction of task variables; the within-area subspace captures the intrinsic within-area interaction between subsets of neurons in either S1 or PPC; the inter-area subspace captures the interaction between S1 and PPC populations, which we could analyze because we simultaneously imaged from S1 and PPC during the task. We aimed to characterize each type of subspace and directly compare them with each other in the same neuronal space. Since generating within-area subspace requires two subpopulations within an area, we randomly split each population into two subpopulations and computed the three types of subspaces from these subpopulations (Fig. 4a). To reduce the

computational complexity and ensure stable model estimation[23,36], we reduced the subpopulation dimensionality with PCA and kept the top 30 PCs prior to the subspace computation. Using more PCs did not change the main findings in the following sections (Supplementary Fig. 4). We repeated this random split procedure 10 times. Because these random splits gave stable results (Supplementary Fig. 5), we averaged the results from all the subpopulations for the following analysis.

We first characterized the encoding subspaces for task variables. We defined the encoding subspace as the axis that captures the difference between the population mean responses to the task variable types[14], for example, texture 1 vs. texture 2 (Fig. 4b). We generated a separate encoding axis for each task variable, using the population activity from the corresponding task window. All encoding axes increased their discrimination index for their target task variable over the course of learning (Fig. 4c), suggesting that the task-related population activity became more separable in this subspace, potentially facilitating the sensory and choice information readout. Individual mice showed variability in these observations that corresponded

to their learning rate; however, averaging sessions for each mouse before pooling the results did not change our observations (Supplementary Fig. 6). Furthermore, the encoding axes from adjacent task windows became better aligned over learning across S1 and PPC areas (Fig. 4d, e), suggesting shared task representation between adjacent windows, consistent with the prolonged responses of task neurons as well as the increase in jointly responsive task neurons (Fig. 2f, g and Supplementary Fig. 3). In PPC-A, the similarity between encoding axes from non-adjacent task windows decreased, suggesting more distinct representations in non-adjacent windows. These results suggest that the encoding subspace reorganized over the learning process to optimize the representation of task information.

### Improved alignment of encoding subspaces with within-area interaction subspaces

We next characterized the within-area interaction subspaces. To do so, we applied canonical correlation analysis (CCA) to the two subpopulations from each area (Fig. 5a). CCA is a statistical method for quantifying the relationship between two sets of variables, and has been used previously to study the interaction between neuronal populations[23,37–39]. Conceptually, CCA finds pairs of projection axes from the two populations that maximize the correlation between the projections (Methods). Similar to PCA, CCA finds uncorrelated sets of projection axes within each population. These axes are ordered with descending correlation values between the paired projections, along the canonical dimensions. The top pair of projection axes thus represents the optimal interaction subspace between the two populations, capturing the strongest canonical correlation.

Given the two subpopulations from the same area, their top canonical correlation value represents the within-area interaction strength, as it reflects the covariance between the subpopulations. Compared to naive sessions, the within-area interaction strength slightly decreased in S1 in expert sessions, whereas it increased in PPC-RL during the texture window and increased in PPC-A during both texture and choice windows (Fig. 5c). These results suggest stronger shared activity within PPC, but diverse subgroups of activity patterns in S1. We further investigated the within-area subspace dimensionality by identifying the number of significant interaction dimensions, defined as the number of canonical dimensions with correlation values higher than the shuffled threshold (Fig. 5b). The shuffled distribution was generated by taking the top canonical correlation from shuffled models, where the trial correspondence was randomized but the overall population firing rate during the task was preserved. The identified significant interaction dimensions captured the instantaneous co-activity between populations. Among the three areas, PPC-A had the highest within-area subspace dimensionality. With learning, the within-area subspaces in S1 and PPC-A significantly increased their dimensionality across task windows, suggesting more coordinated neural activity and consistent encoding patterns, while PPC-RL showed an increase only during the texture window (Fig. 5d). This is consistent with our previous finding that PPC-A but not PPC-RL forms tone-texture associations and generates sensory predictions[23]. We conclude that task-learning expanded the within-area interaction subspace and strengthened the interactions within PPC.

We then wondered whether the within-area interaction axis contained more task-relevant information over the learning process, through improved alignment with the task encoding axis. To answer this question, we computed the cosine similarity between the encoding axis and the top within-area axis computed from the same subpopulation (Fig. 5e), reflecting the alignment between the task-related variability and the within-area population variability. We performed this comparison for each pair of axes from all task windows. In S1, the alignment between the encoding axes and the within-area axes from the texture and choice windows remained stable over the course of

learning. In PPC areas, the texture encoding axis became more aligned with the within-area axes during the texture and choice windows (Fig. 5f), indicating that the within-area activity in PPC during these task windows could carry more texture information, which is essential for the correct choice. In contrast, the tone encoding axis in PPC-A became more distinct from the within-area axes in the following task windows, coinciding with the changes in encoding axis structure. We also quantified the changes in the within-area subspace structure by computing the similarity between pairs of within-area axes, which reflects the similarity of shared activity patterns across task windows. We found that the within-area axis structure across task phases did not show increased alignment in PPC-A, and even became more distinct in S1 and PPC-RL over the learning process (Supplementary Fig. 7a), indicating distinct neuronal patterns across task windows in S1 and PPC-RL. Together, these results suggested an improved alignment between task encoding subspaces and within-area subspaces in PPC during learning, presumably driven by an optimized encoding subspace structure and potentially contributing to improved task representation by the local populations.

To determine if task information in the within-area subspace increased over learning, we projected population activity onto the within-area axes and performed a decoding analysis for each task variable (Fig. 5g). In expert sessions, the discrimination index of within-area activity was overall improved across S1 and PPC areas (Fig. 5h), in agreement with the increased number of task neurons as well as improved population encoding performance. Therefore, task-relevant information was enhanced across within-area subspaces in S1 and PPC during learning.

### Improved alignment of encoding subspaces with inter-area interaction subspaces

Our experimental design of simultaneous imaging from S1 and PPC areas allowed us to also probe the inter-area interaction subspaces between them. To do so, we applied CCA to the subpopulations from S1 and PPC-RL, or from S1 and PPC-A (Fig. 6a). As for the within-area subspace analysis, we computed the top canonical correlation strengths and the number of significant dimensions of interactions. Over the course of learning, the interaction strength between S1 and PPC-RL slightly decreased during the tone and reward windows, while their interaction subspace dimension expanded during the texture window (Fig. 6b). In contrast, S1 and PPC-A interaction strength remained stable and consistently higher compared to S1 and PPC-RL interactions. Additionally, the interaction subspace dimensionality increased for S1 and PPC-A during the tone, texture and choice windows (Fig. 6c), suggesting enhanced coordination and communication, potentially due to the engagement of both populations in a common computational process driven by the common task inputs. This enhanced communication of S1 and PPC-A is also consistent with our previous finding of strong interactions between S1 and PPC-A, but not PPC-RL, during sensory predictions[23].

We then compared the alignment between the inter-area axes and the encoding axes for each pair of areas, measured by the cosine similarity (Fig. 6d). Both PPC-RL and PPC-A showed improved alignment of their encoding axes and their inter-area axes with S1 over the learning process. In PPC-RL, its inter-area axes with S1 after the texture onset became more aligned with the texture and choice encoding axes (Fig. 6e), suggesting more efficient and coordinated information transfer between S1 and PPC-RL. In PPC-A, this improvement in alignment occurred earlier during the trials, prior to texture onset, where the inter-area axes during the tone window developed texture and choice information over learning (Fig. 6f). Such improvement also occurred in the S1 interaction subspace with PPC-A. These results suggest that PPC-A forms sensory associations between tone and texture and plays an important role in multisensory processing in this specific task. To understand whether these

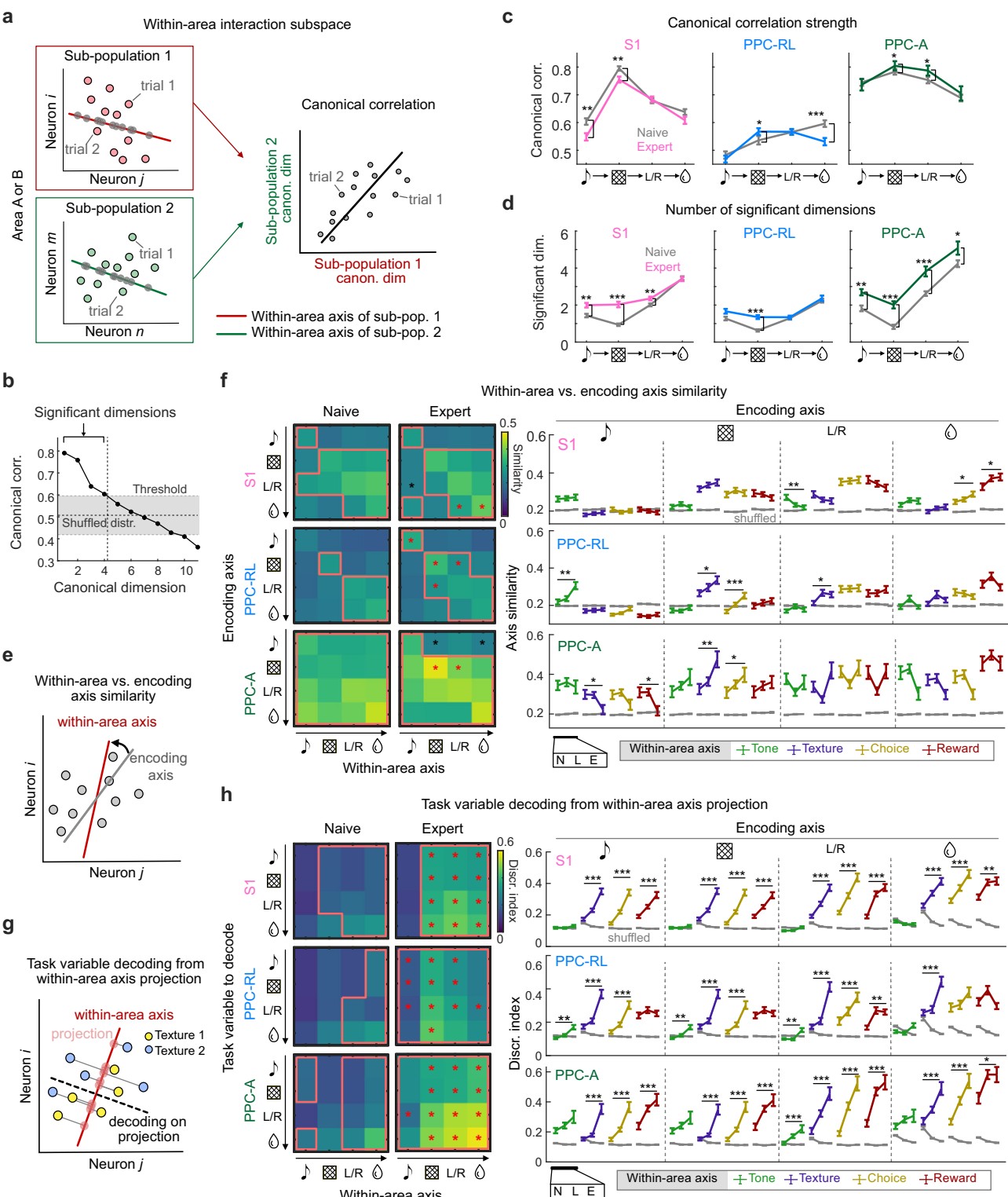

changes in alignment can be explained by the changes in the intrinsic inter-area axis structure, we also analyzed the similarity between pairs of inter-area axes across task windows. We found that the alignment between inter-area interaction axes across task windows did not change significantly overall during learning (Supplementary Fig. 7b, c). These results suggest an improved alignment between task encoding subspaces and S1-PPC interaction subspaces during learning, potentially contributing to improved communication of task information between areas.

We further tested whether the inter-area subspace developed more task information with learning, following its improved alignment with the task encoding subspaces. As described above, we projected population activity onto the inter-area axes and performed a decoding analysis for each task variable (Fig. 6g). We found that the projected inter-area activity of S1 and PPC showed improved discriminability of task variables (Fig. 6h, i). Therefore, the interaction between S1 and PPC areas carried enhanced task-relevant information over the course of learning.

**Fig. 5 | Task encoding axes aligned with PPC within-area axes during learning.**
**a** Diagram of using CCA to generate pairs of within-area interaction axes from sub-populations of the same area, which maximizes the canonical correlation between the projected data from each sub-population. **b** Example of identifying significant CCA dimensions. Black line represents canonical correlation in descending dimensions; gray zone represents confidence interval from shuffled models. **c** Canonical correlation strength in naive and expert condition for each task window, in S1, PPC-RL, and PPC-A. Line colors represent learning phases. **d** Number of significant dimensions in naive and expert condition for each task window, in S1, PPC-RL, and PPC-A. **e** Diagram of comparing within-area interaction axis with encoding axis, using cosine similarity. **f** Left: pairwise similarity between within-area interaction axis (x-axis) and encoding axis (y-axis), across task windows. Red boxes indicate that the data is above the corresponding shuffled distribution; red and black asterisks indicate that the data from the expert condition is significantly

higher or lower than the naive condition. Right: quantification of axis similarity. Line colors represent the task window of the within-area axis; gray lines represent 95% quantile of shuffled data. **g** Diagram of computing decoding performance from projected population activity on the within-area interaction axis. This example shows texture decoding, using neural activity during the texture window to project on the within-area axis. **h** Left: discrimination index of task variables (y-axis), using projection on the within-area axis from each task window (x-axis). Lines and asterisks are represented as in (**f**). Right: quantification of discrimination index. Colored lines represent the task window of the within-area axis. (S1: 13 mice, 122, 171, 119 sessions [naive, learning, expert]; PPC-RL: 13 mice, 66, 72, 86 sessions; PPC-A: 9 mice, 56, 99, 33 sessions; $*p < 0.05$, $**p < 0.01$, $***p < 0.001$; (**f**, **h**) left panels: $*p < 0.05$; two-sided Wilcoxon rank-sum test against naive condition; $p$-values were corrected for multiple comparisons using FDR method; mean ± SEM. Source data are provided as a Source Data file.

## Disrupted task representation and misaligned subspaces during mistakes

During the expert phase, mice still made mistakes (Fig. 1c). In these expert incorrect trials, the licking pattern as well as the pupil diameter and face movement of mice were different from the incorrect trials in naive sessions, but were overall similar with the expert correct trials, despite slightly increased face movement during tone and reduced face movement during choice (Supplementary Fig. 8). We wondered whether these expert incorrect choices can be explained by changes in the underlying task representation on both single-neuron and population levels. We first compared the activity of task neurons during correct and incorrect trials in expert sessions. Compared to correct trials, the response of task-discriminative neurons showed reduced activity strength and shortened response profile in incorrect trials (Fig. 7a, b), suggesting a weakened neural representation. While population dimensionality in the neuron and trial spaces was minimally affected by incorrect behavior during the expert phase (Supplementary Fig. 9a–d), the encoding subspace of S1 and PPC populations in incorrect trials showed reduced task variable discriminability (Fig. 7c). Consistently, the top PCs in neuron space also carried less task information during incorrect trials across S1 and PPC areas (Supplementary Fig. 9e).

Finally, we investigated the changes in within-area and inter-area subspaces during incorrect trials. We generated within-area and inter-area subspaces for correct and incorrect trials separately, using matching number of trials (Methods). While both within-area and inter-area interaction strength and dimensionality were minimally affected by incorrect behavior (Supplementary Fig. 9f–k), these axes were less well aligned with the task-encoding axes compared to correct trials in S1 and PPC-A (Fig. 7d–h), both in their within-area interaction subspace (Fig. 7f) and in their communication subspace (Fig. 7h), while PPC-RL was not affected (Fig. 7f, g). In addition, population activity projected onto both within-area and inter-area axes carried less task information (Supplementary Fig. 10). Together, these results suggest that behavioral mistakes in expert mice were associated with less reliable neuronal responses and misaligned encoding and intrinsic interaction subspaces, potentially leading to erroneous task information processing and behavioral output.

## Discussion

Using simultaneous multi-area two-photon imaging, combined with a sensory discrimination task and longitudinal behavioral and neuronal recordings throughout learning, we provided a systematic overview of learning-related changes in S1 and the higher association areas PPC-A and PPC-RL, both required for optimally performing the task. We discovered changes on different levels, supporting multiple learning mechanisms. On the single-neuron level, additional task neurons were recruited, and the task-related neuronal responses were enhanced. On the population level, neuronal response dimensionality and trial-to-trial variability decreased differentially in S1 and PPC. Task

representation was enhanced through reconfigured task encoding axes. The improved population readout was also accompanied by increased alignment between the task encoding subspaces and the within- and inter-area subspaces in PPC areas. Incorrect choices of expert mice were accompanied by less reliable neuronal responses and misaligned encoding and intrinsic interaction subspaces, potentially contributing to the behavioral mistakes.

At the local population level, task-learning can be accompanied by enhanced task information readout in both primary sensory areas and higher areas[11,12,14,40]. This can be achieved through enhancing neural coding consistency[41] and re-aligning the readout dimensions with the more stable intrinsic within-area subspaces[11]. In our study, we found evidence for both mechanisms. As the task representation improved, the within-area subspaces in PPC also strengthened and expanded, and their alignment with the task encoding dimensions increased. We also observed distinct changes in S1, PPC-RL, and PPC-A that accompanied the improved task representation during learning. S1 and PPC-RL showed decreased neuronal activity dimensionality, decreased within-area axis similarity across time, but stable trial-to-trial variability and increased encoding axis similarity. These findings potentially suggest that the populations compress their variability at each time point to achieve more consistent task encoding, with a dynamic shared variability structure across time. In contrast, PPC-A showed decreased trial-to-trial variability, decreased overall encoding axis similarity, but stable neuronal activity dimensionality and stable within-area axis similarity across time. These adaptations in PPC-A potentially suggest reduced noise levels across trials and diversification of the task-related encoding structure across time, while preserving the variability of the population.

At the multi-area level, enhanced signal propagation between areas could provide another learning mechanism. This can be achieved through stronger shared signals between areas and by better aligning the task information and inter-area communication subspaces[20,42]. For example, studies have shown that task-learning increases shared dimensionality among motor cortex neurons[20] and that top-down signals can change the population interaction structure dynamically during a behavioral task[21]. However, although the inter-area communication subspaces can be flexible[43,44], task-learning does not require reconfiguration of these subspaces. In many cases, the intrinsic within- and across-population activity structures remain stable and impose constraints on task learning[16,22]. Improving task representation along the inter-area subspaces without changing them, e.g., by regulating the internal states such as attention, is sufficient to improve task performance[17,42]. Our results support this mechanism by showing an improved alignment between task encoding subspaces with the S1 and PPC inter-area subspaces, potentially facilitating the readout of relevant task information by PPC.

Task-learning often involves changes in specific pathways[6,18,19,45]. Our study is focused on learning-related changes in the S1 and PPC areas. While both PPC-RL and PPC-A were engaged in this task and required for optimal task performance, learning this task preferentially

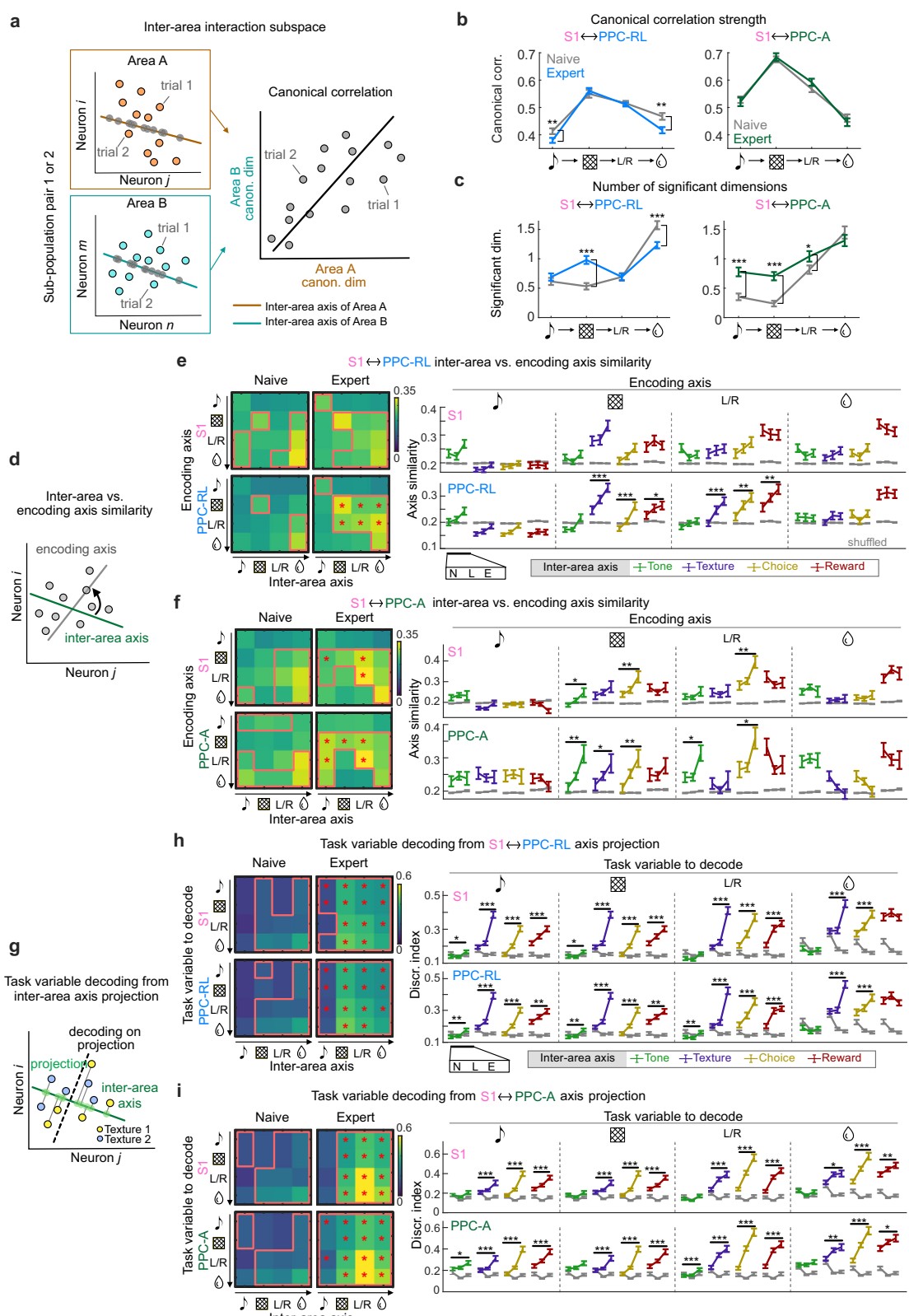

involved PPC-A throughout the tone, texture and choice windows, opening additional within-area interaction subspaces, whereas PPC-RL was involved mostly in texture processing. In addition, the interaction subspaces between S1 and PPC-A expanded throughout the task, whereas S1 and PPC-RL interaction was enhanced only during the texture window. These results agree with our previous findings that PPC-A, but not PPC-RL, encodes predictive texture information using the preceding tone, in expert mice performing the same behavioral

task[23]. As a higher association area, the PPC is important for routing sensory information in behavioral tasks, with its activity emerging during learning as an intermediate step of sensory information flow[26,28]. Our study adds to the accumulating evidence that different PPC subregions have distinct roles[23,26]. The higher engagement of PPC-A compared to PPC-RL in our study is possibly due to the task design, where a specific tone was paired with a fixed subsequent texture, allowing mice to form specific tone-texture associations[23]. In other

**Fig. 6 | Task encoding axes aligned with S1 and PPC inter-area axes during learning. a** Diagram of using CCA to generate pairs of inter-area interaction axes from sub-populations of two different areas (Area A and B), which maximizes the canonical correlation between the projected data from each sub-population. **b** Canonical correlation strength in naive and expert conditions for each task window, for S1 and PPC-RL interaction (left) and S1 and PPC-A interaction (right). Line colors represent learning phases. **c** Number of significant dimensions in naive and expert conditions for each task window. **d** Diagram of comparing inter-area axis with encoding axis. **e** Similarity between inter-area axis and encoding axis for S1 (top) and PPC-RL (bottom) interactions. Left: pairwise similarity between inter-area axis (x-axis) and encoding axis (y-axis), across task windows. Red boxes indicate that the data is above the corresponding shuffled distribution; red or black asterisks indicate that the data from expert condition is significantly higher or lower than the naive condition. Right: quantification of similarity. Line colors represent inter-area axis windows; gray lines represent 95% quantile of shuffled data. **f** Same

plots as in (**e**) but for S1 (top) and PPC-A (bottom) interactions. **g** Diagram of computing decoding performance from projected population activity on the inter-area axis. This example shows texture decoding, using neural activity during the texture window projected on the within-area axis. **h** Task variable decoding from S1 and PPC-RL activity projection on the inter-area axis. Left: discrimination index of task variables (y-axis), using projection onto the inter-area axis from each task window (x-axis). Right: quantification of discrimination index. Colored lines represent inter-area axis windows. **i** Same plots as in (**h**) for task variable decoding from S1 and PPC-A activity projection on the inter-area axis. (S1 and PPC-RL interaction: 13 mice, 66, 72, 86 sessions [naive, learning, expert]; S1 and PPC-A interaction: 9 mice, 56, 99, 33 sessions; *$p < 0.05$, **$p < 0.01$, ***$p < 0.001$; (**f**–**i**) left panels: *$p < 0.05$; two-sided Wilcoxon rank-sum test against naive condition; p-values were corrected for multiple comparisons using FDR method; mean ± SEM. Source data are provided as a Source Data file.

behavioral tasks where only one stimulus type is required for the correct decision, PPC-A has been implicated in auditory discrimination, while PPC-RL was linked to tactile discrimination[26]. However, PPC is a highly flexible area, and changes in the task design could strongly influence whether and how PPC is involved[46].

In our specific task design, there are two types of learning: sensory association learning, which links the tone stimuli to the paired subsequent texture stimuli, and sensorimotor learning, where the tone-texture sequence is associated with a specific choice. We observed some differences regarding these two types of learning. During the tone window, which represents sensory association learning, the populations showed less prominent increases in discrimination ability and number of task-responsive neurons, but more noticeable increases in the population dimensionality as well as the interaction dimensions between S1 and PPC-A. PPC-A to S1 interaction during the tone window also developed information about the upcoming texture, through increased alignment with the texture encoding axis. During the texture window, which is prior to the choice window and thus represents sensorimotor learning, the population showed an increased percentage of task neurons and improved discrimination ability, as well as increased within- and inter-area dimensions. Both PPC-A and PPC-RL improved their texture and choice encoding. We would like to note here that learning a tactile discrimination task is typically accompanied by refined movement patterns[2,3,5,6,30], which could influence the observed neuronal activity. In our case, the neuronal findings were accompanied by changes in facial movement during the tone and texture windows, possibly due to specific whisking patterns that emerged during learning[6,30]. These changes in movement patterns could potentially explain some of the neuronal findings, alongside with the neuronal adaptations underlying sensory processing and decision making related to the task. Together, these results suggest that distinct mechanisms exist for different types of learning. Fully understanding these processes will require further studies.

Overall, our results showed that task-learning is achieved through coordinated changes on multiple levels across neocortical areas. These changes include local recruitment of task neurons, and the improved alignment of task-relevant information with the intrinsic interaction subspaces within and across areas, resulting in improved sensory processing and refined behavioral output. These findings highlight the complexity of learning processes, and we expect them to prompt future research to further understand the specific mechanisms and their impact on neural computation and behavior underlying learning.

## Methods
All procedures of animal experimentation were carried out according to the guidelines of the Veterinary Office of Switzerland and following approval by the Cantonal Veterinary Office in Zurich (licenses 234/2018, 211/2018, 141/2022).

## Mice and dataset
Part of the dataset and method has been previously published[23]. Mice were housed on a 12-h reversed light/dark cycle at an ambient temperature of between 21 °C and 23 °C, with humidity level between 55% and 60%. A total of 15 mice were included in this study. Mice belonged to one of the following transgenic strains: RasGRF2a-dCre;CamK2a-tTA;TITL-GCaMP6f (M10, M11, M12, M26, M28, M29, M33, M34, M35), GP5.17(C57BL/6J-Tg(Thy1-GCaMP6f)GP5.17Dkim/J, Jackson Laboratory 025393) (M14, M15), Snap25-IRES2-Cre-D;CamK2a-tTA;TITL-GCaMP6f (M17), RasGRF2a-dCre;tTA2-GCaMP6f (M30, M38, M40). All mice expressed GCaMP6f in layer 2/3 pyramidal neurons of the neocortex. Both sexes were included in this study (male: M10, M14, M15, M26, M30, M33, M34, M35; female: M11, M12, M13, M17, M28, M29, M38, M40). Among the 15 mice, S1-PPC_A imaging was performed on 9 mice (M26, M28, M29, M30, M33, M34, M35, M38, M40), with 5 mice imaged since naive phase (M33, M34, M35, M38, M40), the rest imaged during late learning and/or expert phase; S1-PPC_RL imaging was performed on 13 mice (M10, M11, M12, M14, M17, M26, M28, M29, M30, M33, M34, M35, M40), with 8 mice imaged since naive phase (M10, M11, M12, M14, M17, M26, M28, M29), the rest imaged during late learning and/or expert phase. Mice were 2.5-4 months old at the beginning of behavior training, and 3-5 months old at the time of imaging.

## Surgical procedures
We performed a craniotomy over S1 and PPC in the left hemisphere of all mice. During surgery, mice were anesthetized with 2% isoflurane mixed with oxygen and maintained at 37 °C body temperature. Mice were treated with analgesia medication (Metacam, 5 mg/kg, s.c.; lidocaine gel over the skull skin) before exposing the skull. Then, a 4-mm round cranial window was made and covered with a glass coverslip using dental cement (Tetric EvoFlow). A light-weighted head-post was mounted on the skull using dental cement. Mice were continually monitored after surgery for at least three days. For strains expressing destabilized Cre (dCre), we induced GCaMP6f expression by intraperitoneally administering trimethoprim (TMP, Sigma T7883; in Dimethyl sulfoxide (DMSO, Sigma 34869) at 100 mg/ml; 150 mg TMP/g body weight) at least one week before imaging started.

## Behavior training
Mice recovered for at least 1 week before the behavior training started. During the first phase of training, mice were handled by the experimenter for several days until showing no sign of stress, then they were gradually accustomed to head fixation. Next, mice were put on water scheduling, and they were introduced to the behavior setup. During the first 2–3 sessions, mice obtained sugar water auto-reward after the choice tone (2 beeps at 3 kHz of 50-ms duration with 50-ms interval), from one of the two lick ports. Once they learned to lick after the choice tone, we introduced the full task.

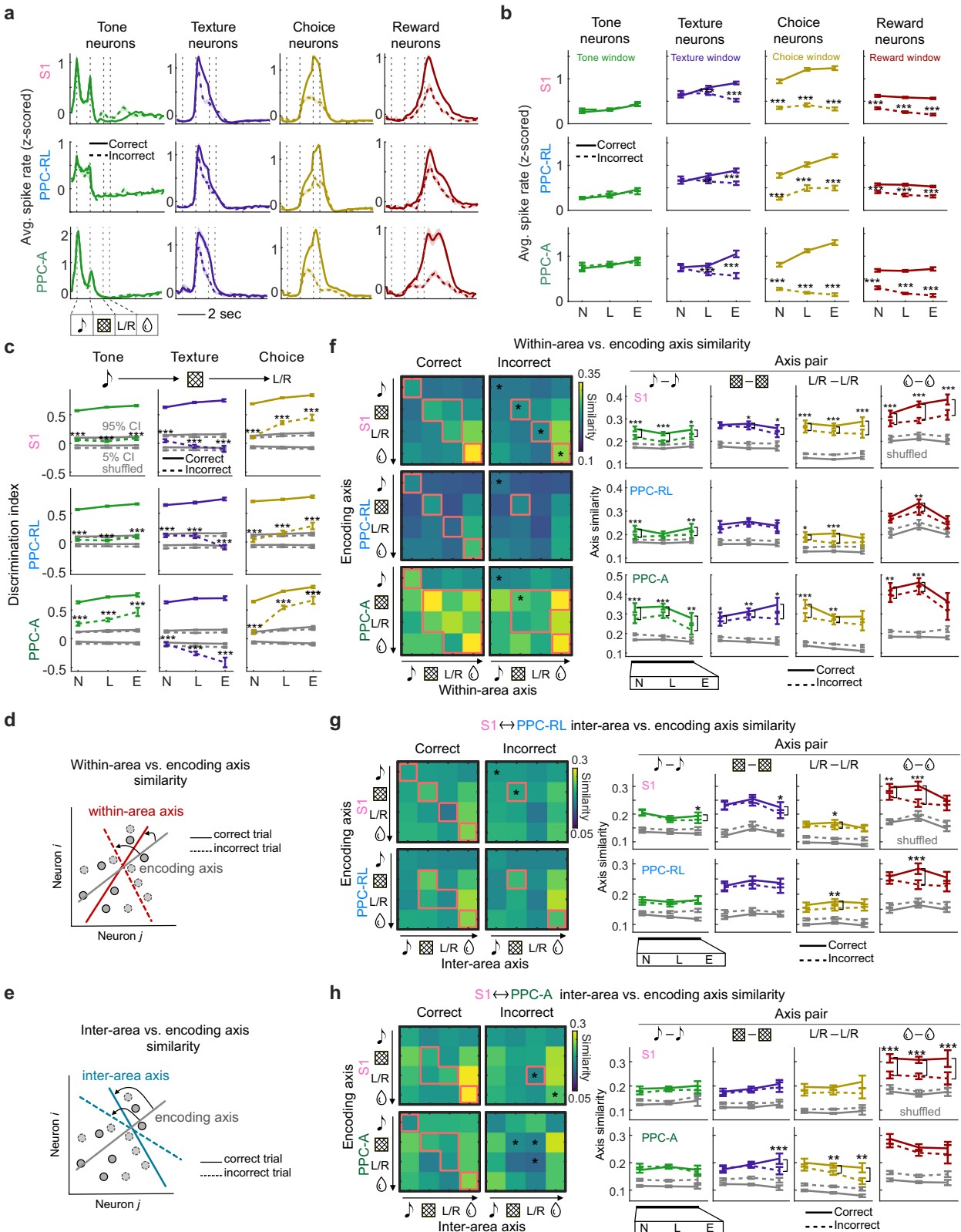

Behavior training was conducted using custom LabView software. Each trial started with either a low tone or a high tone (10 kHz or 18 kHz, 6 repetitions, 50-ms duration and 50-ms intervals), followed by a matched smooth or a rough texture (sandpaper, P100 vs. P1200 for M10–17, P280 vs. P800 for M26–40) carried by a rotary motor, mounted on a linear stage. Both the tone and the texture were presented for 1 s. Then, the choice window started with the choice tone described above, lasting

for up to 2 s. As soon as mice licked, the choice window was terminated, and the reward window started. Each correct choice led to a small sugar water reward (~4 µl) on the respective water spout; incorrect choices were neither rewarded nor punished. The inter-trial interval was randomly distributed between 4–8 s.

Training started with 2–3 sessions of auto-reward, where the reward was automatically delivered from the correct lick port, no matter what

**Fig. 7 | Disrupted task-related neuronal representations during incorrect trials. a** Session-averaged spike rates of task-discriminative neurons during correct (solid lines) and incorrect (dashed lines) trials. Vertical dashed lines indicate task windows. **b** Quantification of average spike rates of task-discriminative neurons during correct (solid lines) and incorrect (dashed lines) trials in corresponding task windows. Colors represent the quantified task window. **c** Discrimination index of population activity projections on the task encoding axes during correct (solid lines) and incorrect (dashed lines) trials in S1, PPC-RL and PPC-A. Gray lines represent shuffled data confidence interval (CI). **d** Diagram of the similarity between the encoding axis and the within-area axes from correct and incorrect trials. **e** Diagram of the similarity between the encoding axis and the inter-area axes from correct and incorrect trials. **f** Similarity between the within-area axis and the encoding axis in expert condition. Left: pairwise similarity between within-area axis (x-axis) and encoding axis (y-axis), in correct (left) and incorrect (right) trials. Red lines indicate that the data is above 95% percentile of the shuffled distribution; red

or black asterisks indicate that the data from expert condition is significantly higher or lower than the naive condition. Right: quantification of similarity. Solid and dashed lines represent correct and incorrect trials, correspondingly; gray lines represent shuffled data. **g** Similarity between inter-area axis and encoding axis in expert condition for S1 (top) and PPC-RL (bottom) interactions. Left: pairwise similarity between inter-area axis (x-axis) and encoding axis (y-axis), in correct (left) and incorrect (right) trials. Right: quantification of similarity. **h** Same plots as in (**g**) but for S1 (top) and PPC-A (bottom) interactions. (S1: 13 mice, 149, 156, 48 sessions [naive, learning, expert]; PPC-RL: 13 mice, 79, 63, 33 sessions; PPC-A: 9 mice, 70, 93, 15 sessions; S1 and PPC-RL interaction: 13 mice, 79, 63, 33 sessions; S1 and PPC-A interaction: 9 mice, 70, 93, 15 sessions; *$p < 0.05$, **$p < 0.01$, ***$p < 0.001$; (**e**, **g**, **h**) left panels: *$p < 0.05$; two-sided Wilcoxon signed rank paired test; $p$-values were corrected for multiple comparisons using FDR; mean ± SEM). Source data are provided as a Source Data file.

the mice chose. Once the mice were accustomed to the task structure, formal training started. During training, we adopted a "repeat incorrect" strategy, where incorrect trials were followed by the same tone-texture stimulus pair until the mouse chose correctly. To motivate the mice, approximately 10% of miss trials were auto-rewarded in the reward window. Each day, the training lasted as long as the mouse was actively engaged in the task, typically 100–500 trials. Mice were trained once per day for 5–6 days a week. Weight, health, and water intake were monitored daily. All training was performed in the dark, with mice continuously monitored with a CMOS infrared-sensitive camera (Basler acA1440–220um) under 850-nm infrared LED background illumination. Behavioral videos were recorded at 50 Hz, and body movement was computed using frame-to-frame correlation. The pupil was constrained by a small UV LED (385 nm, Thorlabs LED385L) positioned close to the eye, and illuminated by the two-photon near-infrared laser. We tracked the pupil diameter by binarizing the pupil image and fitting an ellipse to the pupil region. Body movement and pupil diameter were both smoothed with a median filter of 200-ms width.

To account for the differences in trial number for each day during learning, we split the training days into sessions of 80–120 trials (median: 120 trials; mean ± SD: 116 ± 9 trials) and computed all analysis based on sessions. As this procedure pooled correlated neuronal populations from the same training day as independent samples, we also stratified one session per mouse with the same number of trials for each day, which reproduced our main findings (Supplementary Fig. 11). During the expert phase, mice went through mismatch experiments, where we studied predictive processing, as previously published[23]. Briefly, 10-30% percent of the trials were mismatch trials where tone-texture pairing was inverted. Since the mismatch trials were infrequent and we implemented the repeat incorrect strategy in these experiments to reinforce normal trial pairing, we also included these datasets for analysis, but excluded all mismatch trials as well as sessions with lower performance (<70%).

### Optogenetics
For the optogenetics experiment, we trained 8 VGAT-ChR2-EYFP mice to perform the behavioral task. Before behavior training, we implanted optical fibers (400-µm diameter, NA 0.39) bilaterally above PPC-A and PPC-RL, through a small cranial window. The coordinates used to determine PPC-A and PPC-RL are (−1.8, 2.25) and (−2.5, 3.35), respectively (from Bregma and midline, in mm). Laser light (470 nm, 1 mW above cortex; Thorlabs M470F4) was modulated by a 40-Hz square wave (50% duty cycle) and delivered throughout the texture window for 1 s. In each session, either bilateral PPC-A or bilateral PPC-RL was inhibited in 50% of the trials. Due to the performance fluctuation of this transgenic strain, we excluded the non-expert subsessions during analysis. At the end of all optogenetics sessions, we conducted tone-only and texture-only sessions, where either the tone or the texture were omitted while maintaining the original temporal structure of

trials. Optogenetics inhibition was performed during the texture window, as described above.

### Sensory mapping
The exact locations of S1, PPC-A, and PPC-RL were determined using widefield sensory mapping and retinotopic mapping, as previously reported[23]. Briefly, for widefield mapping, mice were lightly anesthetized and presented with visual, whisker, and hindlimb stimuli (30 repetitions for each modality) on the contralateral side to the imaging window, while we simultaneous performed widefield imaging. In the widefield imaging system, we used a blue LED light source (Thorlabs; M470L3) and an excitation filter (480/40 nm BrightLine HC), a 4x objective (Thorlabs TL4X-SAP, NA 0.2) for imaging, an emission filter (529/24 nm, BrightLine HC), and a CMOS camera (Hamamatsu Orca Flash 4.0) for collection. For retinotopic mapping, a drifting spherically-corrected checkerboard visual stimulus of four cardinal directions (10 repetitions each direction) was presented on an LED screen (Adafruit Qualia 9.7" DisplayPort Monitor, 2048 × 1536 pixel resolution) across the visual field of the mice. The retinotopic map was calculated using a previously reported analysis pipeline[47]. The final locations of S1, PPC-A and PPC-RL were determined by optimally aligning the sensory map and retinotopic map together to the Allen Mouse Common Coordinate[48].

### Two-area two-photon imaging
Two-area two-photon imaging was performed using a previously reported custom-built microscope[24]. The simultaneous two-area imaging was achieved through a temporal multiplexing scheme, where the laser pulses from a Ti: sapphire laser (Mai Tai HP DeepSee, Spectra-Physics) was split in two temporally interleaved copies, each directed through an independently movable unit to a separate field of view. Two-photon imaging was performed at 920-nm excitation with a green emission filter (510/42 nm bandpass), through a 16x objective (N16XLWD, Nikon, NA 0.8). In each area, we performed volumetric imaging using an electrically tunable lens (Optotune EL-10-30-C), from 3 different depths in layer 2/3, separated by 40–50 µm, with typically 450 × 500 µm FOV size, at a resolution of typically 370 × 256 pixels. For two areas and three imaging depths per area, the volume rate was typically 9.3 Hz. During learning, we followed the same FOVs in each mouse, guided by the blood vessels and landmark neurons, with minor shifts each day. In expert experiments, we slightly changed the depths in each session to cover slightly different populations. Due to the dense labeling of the GCaMP6f transgenic mice, the long learning curve, and the shifts in FOVs across days, we could not track enough neurons throughout the whole learning curve, therefore, we did not seek to match the neurons across days.

### Processing of two-photon imaging data
We used Suite2p[31] to perform rigid motion correction on the raw data, model-based background subtraction, neuron identification,

fluorescence extraction, and neuron classification. This pipeline outputs the raw fluorescence traces, the neuropil traces, and the deconvolved spike rates of identified neurons. We manually curated each imaging session and discarded non-neuronal structures or low-quality ROIs. All analysis was performed using the z-scored deconvolved spike rates, where the spike rate of each neuron was normalized to have zero mean and unit standard deviation.

Because fluorescence signal might bleed through between two areas or between two adjacent imaging depths within each area, we carefully removed potentially redundant neurons that were highly correlated with neighboring neurons. We identified potential redundant neuron pairs that fulfilled all the below criteria: (1) spike rate correlation above 0.5; (2) lateral distance between centroids below 5 μm regardless of depths; (3) appeared in adjacent imaging depths in the same imaging area (signal bleed-through in the same area from adjacent imaging planes), or appeared in the same imaging depths in both imaging areas (signal bleed-through across areas from the same imaging plane). In these duplicated neuron pairs, we kept the neuron with the highest average fluorescence level and discarded the one with less fluorescence. In each imaging session, the number of redundant neuron pairs were typically below 5.

Because of the variable length of the choice window, we defined the choice window as the 0.5-s time period before the lick event that triggered the reward window. To exclude lick-related activity during the texture window, we identified all early licks during the texture window and discarded the subsequent activity in these trials until the choice window. We performed this procedure in all analyses except for the sliding window PCA analysis, where discarding early licks during the texture window introduced artificial population dimensionality shrinkage.

### Task neuron analysis

We identified task-tuned neurons by testing the activity level of individual neurons across task windows against a null distribution. We first denoised the spike rates by a small Gaussian kernel (3 frames, sigma = 1); then, for neuron $N_i$ during task window $T_j$, we compared its average activity within the window against a null distribution of average activity level, generated by randomly sampling the same number of frames outside of window $T_j$ for 100 times. The neuron $N_i$ is identified as responsive in task window $T_j$ if its activity in $T_j$ was significantly higher than the null distribution (one-tailed Wilcoxon Rank Sum test; $p < 0.05$). This procedure was performed independently for each session.

To identify task-discriminative neurons, we tested the activity of responsive neurons underlying each task variable state, for tone, texture, choice, and reward. For each task variable (for example, texture), there are two possible values, $s_1$ and $s_2$ (texture 1 and texture 2). For neuron $N_i$, we compared its average activity within the corresponding task window (texture window in this example) between $s_1$ trials and $s_2$ trials. Neuron $N_i$ was identified as a discriminative neuron if its activity was significantly higher (Wilcoxon Rank Sum test, $p < 0.05$) in $s_1$ or $s_2$ trials. We performed this procedure for all four task variables in the corresponding task windows. Responsive and discriminative neurons identified with this method are non-random, as no significant neurons were identified in shuffled data, where the activity of individual neurons was randomized within sessions.

### Discrimination analysis

To calculate the discrimination ability of principal components (PCs) or axis projections, we calculated a standard receiver operating characteristics (ROC) curve using each task variable and computed its area under the curve (AUC). The discrimination index was defined as $DI = (AUC − 0.5) \times 2$.

### Encoding axis analysis

We aimed to compare the encoding axis, the within-area interaction axis, and the inter-area interaction axis within the same space. Since the within-area axis requires two subpopulations from the same area, we generated all three types of axes from the subpopulations. Briefly, each population from S1, PPC-RL, and PPC-A was randomly divided into two subpopulations with a similar number of neurons. Then, to reduce the computational challenge and to ensure a stable CCA computation, we reduced the dimensionality of each subpopulation with PCA. CCA requires a sufficient amount of samples to generate stable solutions. Our dataset consisted of ~120 trials per session, and ~10 time points for each task window. A previous study using simulated datasets has shown that ~50 samples per variable are required to generate a stable solution[36]. To ensure such a condition is met, and to ensure that each subpopulation had a similar number of variables, especially for the inter-area axis computation, we kept the top 30 PCs for each area. The top 30 PCs explained the following fractions of total variance (in %): 41.4 ± 2.0 (naive, mean ± SEM), 41.7 ± 1.5 (learning), 51.7 ± 2.2 (expert, same order for the following) in S1, 37.0 ± 2.6, 38.5 ± 2.6, 47.1 ± 2.5 in PPC-RL, and 38.0 ± 1.6, 42.7 ± 1.4, 50.8 ± 2.9 in PPC-A. Taking the top 60 PCs did not change our main results (Supplementary Fig. 4; explained variance: 63.6 ± 2.0, 65.6 ± 1.7, 75.1 ± 2.1 in S1, 58.5 ± 2.9, 60.3 ± 2.7, 72.0 ± 2.5 in PPC-RL, 62.2 ± 2.2, 68.3 ± 1.7, 77.6 ± 2.8 in PPC-A). For this analysis and all following sections, only datasets with good imaging quality in both simultaneously imaged areas were included. We then computed the three subspaces using each subpopulation. We repeated the random split 10 times, resulting in 20 subpopulations for each area. All results were obtained by averaging these repeats. Only datasets with good imaging quality from both areas, by manual curation, were included in this analysis.

To compute the encoding axis, we calculated the trial average vector for each task variable (for example, texture 1 versus texture 2 trials), and defined the encoding axis as the difference between these two mean vectors. To generate a null distribution of the discrimination index, we shuffled the neuronal ordering of the population activity and computed the discrimination index using the encoding axis from the real data. This shuffling procedure disrupted the contribution of individual neurons to task encoding, while preserving the overall firing rate of the population during the task. To compute axis similarity, in order to account for the differences in the variance captured by each PC, we computed a weighted cosine similarity, defined as $\sum_i w_i u_i v_i / \sqrt{\sum_i w_i u_i^2} \sqrt{\sum_i w_i v_i^2}$, where $u$ and $v$ are the axes, $i$ represents the element in the axis vector, and $w$ is the variance explained by the corresponding PC of each element in the axis vector.

### Within-area and inter-area axis analysis

We utilized canonical correlation analysis (CCA) to evaluate the interaction between neuronal populations within and across different areas[23]. CCA works by identifying pairs of dimensions from the neuronal populations in two neuronal populations, optimizing the correlation between their projected activities. For example, given two neuronal populations whose activity is represented by an $n_x \times t$ matrix **X** from the first population and a $n_y \times t$ matrix **Y** from the second population—where $t$ represents the number of time points and $n_x$ and $n_y$ are the number of neurons in each respective area—CCA determines $\min(n_x, n_y)$ pairs of dimensions, with the correlation between the projections onto these dimensions decreasing from the first pair to the last. Unlike PCA, which independently maximizes the variance explained by the top axes from **X** and **Y**, CCA focuses on maximizing the correlation between the projections of the activity matrices **X** and **Y** onto the identified dimensions.

We performed CCA analysis for the population activity during each task window, where frames pooled across trials were treated as observations. For within-area interaction axes, we performed CCA using the two subpopulations from the same area. For inter-area interaction axes, we performed CCA using two pairs of subpopulations from the two simultaneously imaged areas. To identify interaction

subspaces that reflect the instantaneous co-fluctuation between two populations, we computed 100 shuffled models by randomizing the trial correspondence between the two populations, while preserving the temporal structure of neuronal activity during the task. These shuffled models captured the interaction subspaces resulting from the change in activity levels due to task factors (sensory inputs, choice, reward), but not from the simultaneous interaction between populations. The highest correlation values from these shuffled models represent the shared activity that can be explained by the task structure. We determined the number of significant interaction dimensions using a threshold defined as mean + 1.96 S.D. (standard deviation) of the highest shuffled correlation from the 100 shuffled models. In cases where no significant interaction dimensions were found, the top CCA axes still represent the optimal interaction subspaces, although these subspaces do not contain more information than the effect of common task inputs. Subsequent analysis was performed by always taking the top axes. Using these shuffled models, we generate a null distribution of axis similarity by computing the weighted cosine similarity with encoding axes. Similarity values from the real models were considered significant if they were higher than the 95% quantile of the shuffled distribution. Subsequent statistical comparisons between naive and expert conditions were restrained to pairs where at least one side was significantly above shuffled distribution.

It is worth noting that PCA of the neuron space also captures features of within-area interactions, similar to CCA performed on two subpopulations from an area. However, these two methods reflect different characteristics of the data, with PCA dimensionality capturing the total variability structure during the task, including both shared and independent variability, while CCA dimensionality capturing the reliable shared variability between subpopulations. When the task-relevant variability dominates the total variance and is shared consistently across neurons, both methods define similar axes.

### Analysis of correct and incorrect trials

To account for the differences in trial numbers and to ensure enough samples from each trial type over learning (e.g., expert sessions contain more correct trials than incorrect trials), we re-defined sessions in each imaging day to include 50 trials from each trial type (correct and incorrect trials). For each session, we randomly downsampled the trial type with more trials to generate matching numbers of trials for each trial type. We excluded sessions with less than 30 trials of either trial type to ensure stable analysis results (final session length: median 50 trials for each trial type, mean ± SD: 46 ± 6 trials). We repeated this random downsampling procedure 10 times for each dataset and averaged the results. This procedure was implemented for all analysis concerning the comparison between correct and incorrect trials.

### Statistics and reproducibility

All statistical analysis was done in MATLAB. In general, Wilcoxon signed-rank test was used for paired samples, and Wilcoxon Rank Sum test was used for non-paired samples. No normality test was performed since these tests do not assume normality. Two-sided tests were performed unless otherwise indicated. No statistical methods were used to pre-determine sample sizes, but our sample sizes are similar to those reported in previous publications[1,6,8,26,28,41]. Error bars represent mean ± SEM. $P$-values were corrected for false discovery rate (FDR) < 0.05 for multiple testing, using MATLAB function mafdr. Mice were randomly assigned to imaging groups (S1 and PPC-A or S1 and PPC-RL imaging). Stimulus presentation during imaging was fully randomized. Data collection and analysis were not performed blind to the conditions of the experiments.

### Reporting summary

Further information on research design is available in the Nature Portfolio Reporting Summary linked to this article.

## Data availability

Data are available from the corresponding authors upon request. Source data are provided with this paper.

## Code availability

Data processing and analysis code is available at: https://github.com/HelmchenLabSoftware/multiarea_learning_manuscript.

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

## Acknowledgments

We thank Philipp Bethge for managing the transgenic mouse lines, Fabian Voigt and Hansjörg Kasper for help with optics, and Martin Wieckhorst for the behavior training software. We also thank Matthias Tsai and Christopher Lewis for their feedback on the manuscript. This work was supported by a Sinergia grant from the Swiss National Science Foundation (CRSII5_180316; to F.H.) and a UZH Forschungskredit grant (K-41220-07-01; to S.H.). S.H. received an Ambizione Grant from the Swiss National Science Foundation (PZ00P3_216312). S.H. and F.H. received funding from the University Research Priority Program (URPP) "Adaptive Brain Circuits in Development and Learning" (AdaBD).

## Author contributions

S.H. and F.H. conceived the study and designed the experiments. S.H. performed the experiments and analyzed the data. S.H. and F.H. wrote the manuscript.

## Competing interests

The authors declare no competing interests.
