## [Peer Review file · Nature Communications]

Coordinated multi-level adaptations across neocortical areas during task learning

Corresponding Author: Dr Shuting Han

Version 0:

Reviewer comments:

Reviewer #1

(Remarks to the Author)

The study explores how neural activity evolves during learning, using multi-area two-photon calcium imaging in mouse neocortex during a sensory discrimination task. The authors observed coordinated adaptations in the primary somatosensory cortex (S1) and the anterior (A) and rostralateral (RL) regions of the posterior parietal cortex (PPC). They reported an increase in the number and stability of task-related neurons at the single-neuron level, and at the population level, a rise in dimensionality and reduced variability, enhancing task information encoding. The authors emphasize that area PPC-A becomes more involved during learning, forming novel functional subspaces that align with task encoding.

While I appreciate the authors' focus on learning-related activity changes and cross-area interactions, both of which are timely and important topics, I have reservations about the strength of their conclusions. I am unconvinced that PPC-A plays a remarkable role in learning compared to other regions. The effect sizes for learning-related changes in PPC-A do not appear particularly pronounced, and the optogenetic inhibition experiments (Sup Fig 1) fail to distinguish PPC-A from PPC-RL in a meaningful way. There are multiple qualitative statements asserting PPC-A's unique role (e.g., "particularly in PPC-A"), yet these claims lack strong statistical backing. I recommend providing clear quantitative evidence to support the proposed distinctions.

Additionally, while the optogenetic inhibition data suggests both PPC-A and PPC-RL are comparably important for task performance post-learning, the authors do not present data showing whether these areas contribute during the learning phase itself. Clarifying whether PPC-A plays a more significant role than PPC-RL during learning would strengthen the argument.

Another concern relates to the potential influence of orofacial movements on neural activity and interaction measures. The authors report small but significant changes in facial and licking behavior during learning and across correct vs. incorrect trials. It is crucial to demonstrate that these movements do not confound neural activity or the interaction metrics. Furthermore, do optogenetic inhibitions of PPC-A and PPC-RL impact whisking or licking behavior? Addressing these points would help clarify the interpretation of the neural findings.

Finally, the current claim that PPC-A is more critical for tactile discrimination than PPC-RL appears inconsistent with the authors' prior study (Gallero-Salas et al., Neuron 2020). In that work, PPC-A was implicated in auditory discrimination, while PPC-RL was linked to tactile discrimination. While the behavioral paradigms differ (delayed go/no-go vs. 2AFC with no delay), this alone does not fully account for the apparent contradiction. The authors should reconcile these discrepancies.

Minor comments:

Several figure panels are difficult to read due to their small size, particularly when trying to identify which comparisons are marked by asterisks. This is especially evident in Figs. 6f-i (right panels) and 7e, g, h (right panels), where the details are challenging to see. Enlarging these panels or providing clearer indications of the statistical comparisons would improve readability.

(Remarks on code availability)

Reviewer #2

(Remarks to the Author)

This paper studies how neural activity changes in somatosensory area S1 and two areas of posterior parietal cortex (PPC-A and PPC-RL) during auditory and texture learning in mice. The authors study how neural activity within each of the areas, and how the interactions between areas, changes with learning. Furthermore, they study how the strength and axes of task encoding (including tone, texture, choice, reward) in the neural activity change with learning, as well as how these axes align with the within-area and inter-area activity spaces.

This paper brings together state-of-the-art optical imaging methods for recording the activity of many neurons in multiple brain areas simultaneously while mice are engaged in an interesting learning task. Furthermore, they study the activity using population analysis methods (such as PCA and CCA), which recent studies have used to gain deeper insight into the neural activity than what is possible by studying each neuron individually.

As much as I wanted to like this paper, it was a (very) difficult read, for the reasons described below. I felt overwhelmed with the number of comparisons being made, and as a result it was difficult to come away with a clear message of the findings. The authors try to summarize the findings, but the takeaways provided by the authors are too simple for what the data show. This paper reads like a series of exploratory analyses without clear hypotheses or take-away messages that are i) well-reasoned in the context of the brain areas (S1, PPC-A, PPC-RL) and learning task being studied, and ii) supported by the data.

Specific comments:

1) Throughout the paper, the authors do a large number of statistical tests and just focus on the comparisons that are statistically significant. The authors try to summarize all the comparisons with a sweeping statement, but in each case the statement is far too simple for what the data show. For example, in Fig. 5f, is there a principled reason by which we can understand why PPC-A would significantly increase for only the Texture and Choice curves within the Texture period? What does it mean that the Texture and Reward curves *decrease* during the Tone period? What about all the other comparisons that are not significant - are we allowed to ignore them? How reliable are these effects across individual mice? The same types of questions arise throughout the manuscript, including in Figures 4e, 5c, 6b, 6c, 6f, 6g, 7e, 7g, 7h, Supplemental Fig. 4, and Supplemental Fig. 5.

2) The axis similarity metric, which is used throughout in the paper (e.g., Fig. 4e, Supplementary Figure 4), is difficult to interpret. If this metric changes from novice to expert, it could either be that one axis changes or both axes change. Conversely, if this metric is unchanged from novice to expert, it still could be that both axes change (but the similarity between them remains unchanged). For this reason, the following key claims are not supported by the current analyses:

L247: "We found that ...remained stable in PPC-A"

L249-251: "these results suggest that...during learning"

L276: "mostly remained stable over the learning process"

3) The authors need to be clearer about how they ruled out alternative explanations of their results. For example:

a) In several analyses, the authors perform a shuffle analysis as a control. In each case, they should carefully explain what the shuffle is designed to do, and justify why it is the right control for that analysis.

b) Can some of the results (e.g., Fig 4d,e, Fig. 5f, Fig. 6) be solely explained by increases in firing rates during learning?

c) Do the results in Fig. 5f follow trivially from the results in Fig. 3g because the identified within-area axis structure is sensitive to task encoding?

d) Are the number of neurons being analyzed matched over time (novice vs. expert) within each brain area? How do differences in the number of neurons recorded in each area influence the results?

4) Similar within-area questions are asked using different methods in Fig. 3 (PCA) and Fig. 5 (CCA, by arbitrarily dividing the population into two subsets), making the resulting analyses conceptually redundant. How does one reconcile the sometimes different results obtained by the two methods? For example:

- reconciling Fig. 3c (middle) with Fig. 5d (middle), which look inconsistent with each other.

- relating Fig. 3g to Fig. 5h.

5) I tripped in a large number of locations, which made this paper difficult read. Here are some examples of where I tripped:

L130-131: the logic is unclear why this suggests "both areas are involved in tone-texture association".

L151-152: "PPC-A developed a higher percentage..." How is this statement supported by what is shown in Fig. 2d?

L153: "showed a similar trend" There are so many differences between Fig. 2e and Fig. 2d that are being ignored by the authors.

Fig 2f: is the definition of each type of neuron based on the naive or expert period? Is the definition based on if the neuron is task-responsive or task-discriminative?

Fig 3b: Is the vertical axis the number of dimensions or a percentage?

Fig 3d-f: what is the logic of analyzing activity in this trial space (where each dot is neuron)? If the goal is to assess trial-to-trial variability, why not analyze it in the neuron space, as is standard in our field?

L184: "...through redundant representation" - I don't understand this logic.

L213: should this read "LESS distinct from each other"?

L221: CCA finds uncorrelated dimensions, not orthogonal dimensions.

L227-229: how to interpret these observations?

L232: why take only the top canonical dimension?

L234: why does higher-order area imply higher dimensionality? this is not obvious.

L237: "expanded and enhanced" - what does this mean?

L247: "became more distinct" - what does this mean, and how should the reader interpret these changes?

L261-267: are these results what the authors would have predicted? how do we understand these results? I'm having a hard time wrapping my mind around what's increasing, what's decreasing, and what's staying the same.

Fig 6c: for dimensionalities less than 1, does it mean that the authors are consistently finding that there are no across-area dimensions above chance? How are the subsequent analyses performed in this case?

Other questions:

L200: what is the percentage of variance retained by the top 30 PCs? Is this preprocessing step capturing almost all of the variance in the population activity, as this could influence the CCA dimensions found?

How similar are each of the axes (encoding axes, within-area axes, inter-area axes) to the [1,1,1...] direction? This would inform whether the identified axes are driven by general increases in firing rates across the population.

(Remarks on code availability)

Reviewer #3

(Remarks to the Author)

The authors investigate the neuronal population dynamics that accompany and potentially support learning of a sensory discrimination task. They find that task learning is marked by an increased number and stabilized responses of individual task neurons as well as increased dimensionality and reduced trial-to-trial variability of population responses. Further, task learning is accompanied by changes of within-area as well as between-area subspaces. Most markedly, the authors demonstrate that task encoding and cross-area interaction subspaces are related. Finally, behavioral errors go along with decreased neuronal responses, decreased encoding accuracy and misaligned subspaces.

As far as I know the literature, I think the present work adds substantial evidence to the growing body of knowledge on how neuronal populations support cognition in general and during learning in particular. I find the multi-level description presented ranging from encoding properties of single neurons and local populations to cross-area interactions between populations to be encompassing and illustrative. Last but not least, the investigation of how information coding by populations of neurons as well as interactions between populations, both within as well as across areas, are related is very noteworthy.

As far as I can judge, in general the research presented here is sound. The last author is an eminent figure in the field of two-photon imaging and parts of the data have been recently published in a high-impact, peer-reviewed journal (Han and Helmchen, *Nat Neurosci*, 2024). Data analysis mostly appears to be sound, with methods used being well established in the field and applied appropriately.

That said, I have a few comments/questions/concerns regarding aspects of the data analysis:

1. Mice performed different numbers of trials each day. As the authors write, "To account for the different trial numbers each day, we split each training day into sessions of 80-120 trials." I'm wondering how this procedure affects the analysis outcome, since in this way data from the same training session will be mixed with data from another session. Thus, data samples are not equally independent which will affect the variance of the data. Why didn't the authors simply stratify trial numbers across training days, sampling from the daily outcome of 100-500 trials that they had (taking e.g. $n=100$ trials per day)? My best guess is that this won't badly affect the most robust findings of their study. I hence would like to ask for a re-analysis of the data after subsampling and stratification across training days.
2. In Figure 2, changes in the percentages of task-responsive and task-discriminative neurons are compared. It is not mentioned anywhere that one can expect a certain number (percentage) of neurons to be responsive to task variables (and hence to differentiate between them) or not just by chance. Please make that clear in the text and indicate the related significance threshold assumed (5%, say) in the figures.
3. What was the reason to select precisely 30 PCs of the subpopulation dimensionality before subspace computation?
4. In Figures 5-7, image plots of data matrices are shown including statistics for tests of "real" data against a shuffle distribution (red boxes) as well as asterisks indicating a significantly higher value in comparison with the respective contrasted condition (e.g., naïve vs. expert). In some of these matrix fields, there are asterisks indicating a significantly higher or lower value while at the same time, neither is surrounded by a red box. Does that mean that neither condition has a value higher or lower than the shuffle distribution but when compared against each other, one is significantly higher than the other? But then, what is being compared here; merely noise? Please clarify this and only consider as valid those comparisons where at least one side of a pair is significant as compared to the shuffle.

The above comments/concerns/requests notwithstanding, the present work mostly supports the conclusions provided and the claims raised, requiring no further basic evidence. However, I would like the discussion to be more focused on the truly novel aspects of the results and how the study expands our current understanding of how neuronal population dynamics shape learning processes.

Taken together, I in principle endorse the publication of the present work in *Nature Communications* and thank the authors for the effort they put into it. Please adequately address the questions/concerns raised above beforehand.

(Remarks on code availability)

Version 1:

Reviewer comments:

Reviewer #1

(Remarks to the Author)

I appreciate the authors' revisions in response to my comments as well as those from other reviewers.

1) However, I find the authors' claim that anticipatory whisking or facial movement is inherently part of task learning to be somewhat rough and overly broad, especially considering that "task learning" in this study appears to focus on cognitive task learning—particularly processes related to "sensory processing and decision-making" as stated in the abstract.

The authors state: "We still observed increased facial movement during the texture window even without early licking frames; however, this is likely due to the enhanced and anticipatory whisking that emerges during learning ... This change in whisking pattern is part of task learning and thus is not in conflict with our neural findings but rather could serve as one mechanism that explains certain neural findings."

This response conflates general behavioral adaptation with task-specific cognitive learning while failing to carefully distinguish between them. If any behavioral change occurring during training is categorized as "task learning," it raises the question of why the task was designed to require specific cognitive components in the first place. Under such a broad definition, there is a risk that some of the observed neural changes may reflect epiphenomena related to motor adaptation rather than the cognitive processes that the task was intended to assess.

To be clear, I am not suggesting that all observed neural changes and inter-regional dynamics can be attributed to movement-related factors. However, the authors should at least explicitly acknowledge that motor coordination, alongside task-related sensory processing and decision-making, could influence the neural findings. Clarifying this distinction would strengthen the interpretation of their results.

2) I also appreciate the authors sharing the optogenetic silencing results of mismatched trials in expert mice as Extra Fig. 1. I agree that, while informative, this analysis is not essential to the current study's focus on learning-related changes, and its inclusion is not necessary.

3) Although the authors mentioned that they have redesigned Figures 5–7, many subpanels are still too small, significantly reducing clarity. If the goal is to effectively present these data to readers, space should not be a limiting factor.

Reviewer #3

(Remarks to the Author)

I appreciate the effort the authors put into the revised version of the manuscript. From my side, almost all the issues I've raised have been addressed to my satisfaction.

There is one aspect though that escaped my attention during the first round and that I now noticed. In Figure 2 panels d and e (lines 147-156), the authors report the percentages of neurons being responsive to task variables and task-discriminative. They then go on to make statements about percentages of responsive and discriminative neurons changing over the course of learning. They provide non-parametric test statistics to underscore their claims, but I don't think that this is the right statistic here. As far as I can see, a Chi-square or similar test statistic would be needed to compare percentages of responsive or discriminative neurons here. Please comment on this and provide the appropriate analyses and results to back up the original claims.

Thanks much again for your efforts!

Version 2:

Reviewer comments:

Reviewer #1

(Remarks to the Author)

I am satisfied with the revision and have no further comments.

Reviewer #3

(Remarks to the Author)

Dear editors,

All of the concerns I raised in the previous round of reviewing the manuscript by Han and Helmchen have been adequately addressed by the authors. I thus support publication of the manuscript in Nature Communications.

Thanks and best regards,
Constantin von Nicolai

Response to the Reviewers' comments, Han and Helmchen, manuscript NCOMMS-24-59079-T

Dear Reviewers,

Thank you for your review of our manuscript “Coordinated multi-level adaptations across neocortical areas during task learning”. We appreciate your effort in reviewing our manuscript. We have performed additional analysis and revised our manuscript to address your comments and concerns. By taking into account your inputs, we believe the manuscript is now more rigorous and more clearly presented.

In the following, we list the original comments from the reviewers, followed by our responses (in blue), and how we have addressed the points in the revised manuscript (in orange). We appreciate your evaluation of our responses and the revised manuscript. Besides this letter and a clean version of the revised manuscript, we also attach a version with tracked changes (changes highlighted in red) for your convenience.

Thank you again for your time and effort.

Shuting Han and Fritjof Helmchen

Zurich, 25 February 2025

REVIEWER COMMENTS

Reviewer #1 (Remarks to the Author):

The study explores how neural activity evolves during learning, using multi-area two-photon calcium imaging in mouse neocortex during a sensory discrimination task. The authors observed coordinated adaptations in the primary somatosensory cortex (S1) and the anterior (A) and rostralateral (RL) regions of the posterior parietal cortex (PPC). They reported an increase in the number and stability of task-related neurons at the single-neuron level, and at the population level, a rise in dimensionality and reduced variability, enhancing task information encoding. The authors emphasize that area PPC-A becomes more involved during learning, forming novel functional subspaces that align with task encoding.

While I appreciate the authors' focus on learning-related activity changes and cross-areal interactions, both of which are timely and important topics, I have reservations about the strength of their conclusions. I am unconvinced that PPC-A plays a remarkable role in learning compared to other regions. The effect sizes for learning-related changes in PPC-A do not appear particularly pronounced, and the optogenetic inhibition experiments (Sup Fig 1) fail to distinguish PPC-A from PPC-RL in a meaningful way. There are multiple qualitative statements asserting PPC-A's unique role (e.g., "particularly in PPC-A"), yet these claims lack strong statistical backing. I recommend providing clear quantitative evidence to support the proposed distinctions.

Response:

Thank you for evaluating our manuscript and for your suggestions. We would like to emphasize that despite some differences between the PPC areas, our main observation is that both PPC-A and PPC-RL were involved and required for this task (as also shown by the optogenetic experiments; see additional details in our response to the next point). The main conclusion of this manuscript thus is that both S1 and PPC experienced learning-related multi-level changes in a coordinated way, supporting learning mechanisms on different scales.

Given the rich details in the dataset and the complexity of analysis on multiple levels, we have focused on characterizing learning-related changes by comparing naive condition with expert condition. We have hence revised our manuscript to highlight these learning-related changes. In some cases, we noticed that PPC-A and PPC-RL showed some differences in these changes. These differences are noticeable in Fig. 2d-e (PPC-A showed more task-related neurons than PPC-RL), Fig. 5d (PPC-A has higher within-area subspace dimensionality than PPC-RL), Fig. 6c (PPC-A has higher interaction subspace dimensionality with S1 than PPC-RL), and Fig. 6e-f (PPC-A interaction subspace during tone window aligned with task encoding axis, but not PPC-RL). In these cases, we now provide qualitative statements to describe these differences in our revised manuscript, without direct comparison between areas. We have refrained from introducing additional statistical comparisons between areas to keep the flow and focus of this manuscript. Throughout the manuscript, we removed the "particularly in PPC-A" side remark.

We would like to indicate that the reason why we initially highlighted the differences between PPC-A and PPC-RL relates to our previous study (Han and Helmchen, 2024). In this study, using the same behavioral task with additional mismatch trials, we observed that PPC-A but not PPC-RL formed tone-texture association and generated predictions of the texture using tone information. This study was performed in expert condition. Given these findings that indicated different roles of PPC-A and PPC-RL in this task, we complimented the expert results with learning-related differences in this current study. In the revised

manuscript, we have included more information about our previous study to support our motivation in comparing the results between PPC-A and PPC-RL.

Specific changes in the revised manuscript:

- We revised our statements to highlight the general differences observed between naive and expert states. We also removed statements that involves direct comparison between PPC-A and PPC-RL. These revisions include:

“The PPC areas were gradually engaged during the learning process, expanding their within-area and inter-area interaction subspaces with S1.” [Removed “particularly PPC-A”] (Line 70-71)

“Thus, task-learning was accompanied by an overall recruitment of task neurons as well as extended and enhanced neuronal responses.” [Removed “particularly in S1 and PPC-A”] (Line 161-162)

“Compared to naive sessions, the within-area interaction strength slightly decreased in S1 in expert sessions, but increased in PPC-RL during the texture window, and increased in PPC-A during both texture and choice windows (Fig. 5c).” [Removed “particularly PPC-A”] (Line 233-235)

“We conclude that task-learning expanded the within-area interaction subspace and strengthened the interactions within PPC.” [Removed “particularly PPC-A”] (Line 247-248)

“In PPC areas, the texture encoding axis became more aligned with the within-area axes during the texture and choice windows (Fig. 5f)” [Removed “particularly PPC-A”] (Line 255-256)

“In expert sessions, the discrimination index of within-area activity was overall improved across S1 and PPC areas” [Removed “particularly PPC-A”] (Line 269-270)

“We found that the projected inter-area activity of S1 and PPC showed improved discriminability of task variables (Fig. 6h-i)” [Removed “particularly for S1 and PPC-A interaction”] (Line 304-305)

“these axes were less aligned with the task-encoding axes compared to correct trials in S1 and PPC-A (Fig. 7d-h), both in their within-area interaction subspace (Fig. 7f) and in their communication subspace (Fig. 7h), while PPC-RL was not affected (Fig. 7f,g).” [Removed “particularly PPC-A” and added “while PPC-RL was not affected”] (Line 325-327)

“In addition, population activity projected onto both within-area and inter-area axes carried less task information (Supplemental Fig. 10).” [Removed “particularly for the inter-area axes of S1 and PPC-A”] (Line 328-329)

“The improved population readout was also accompanied by increased alignment between the task encoding subspaces and the within- and inter-area subspaces in PPC areas” [Replacing “Furthermore, learning this task preferentially involved PPC-A throughout the trial structure, opening additional interaction subspaces within PPC-A and between S1 and PPC-A, while PPC-RL was involved mostly in texture processing.”] (Line 341-343)

“As the task representation improved, the within-area subspaces in PPC also strengthened and expanded, and their alignment with the task encoding dimensions increased” [Removed “particularly PPC-A” and rewrote the statement] (Line 349-351)

- We added more information regarding our previous study to support the observed differences between PPC-A and PPC-RL:

“This is consistent with our previous study that PPC-A but not PPC-RL forms tone-texture association and generates sensory predictions²³.” (Line 246-247)

“Additionally, the interaction subspace dimensionality increased for S1 and PPC-A during the tone, texture and choice windows (Fig. 6c), suggesting enhanced coordination and communication, potentially due to the engagement of both populations in a common computational process driven by the common task inputs. This enhanced communication of S1 and PPC-A is also consistent with our previous finding of strong interactions between S1 and PPC-A, but not PPC-RL, during sensory predictions²³” (Line 282-286)

“In PPC-A, this improvement in alignment occurred earlier during the trials, prior to texture onset, where the inter-area axes during tone window developed texture and choice information over learning (Fig. 6f). Such improvement also occurred in S1 interaction subspace with PPC-A. These results suggest that PPC-A forms sensory association between tone and texture, and plays an important role in multisensory processing in this specific task.” (Line 291-295)

“These results agree with our previous findings that PPC-A, but not PPC-RL, encodes predictive texture information using the preceding tone, in expert mice performing the same behavioral task²³” (Line 376-378)

- We also summarized the differences we observed in S1, PPC-RL and PPC-A in the discussion section, and discussed their implications:

“As the task representation improved, the within-area subspaces in PPC also strengthened and expanded, and their alignment with the task encoding dimensions increased. We also observed distinct changes in S1, PPC-RL, and PPC-A that accompanied the improved task representation during learning. S1 and PPC-RL showed decreased neuronal activity dimensionality, decreased within-area axis similarity across time, but stable trial-to-trial variability and increased encoding axis similarity. These findings potentially suggest that the populations compress their variability at each time point to achieve more consistent task encoding, with a dynamic shared variability structure across time. In contrast, PPC-A showed decreased trial-to-trial variability, decreased overall encoding axis similarity, but stable neuronal activity dimensionality and stable within-area axis similarity across time. These adaptations in PPC-A potentially suggest reduced noise levels across trials and diversification of the task-related encoding structure across time, while preserving the variability of the population.” (Line 349-359)

“While both PPC-RL and PPC-A were engaged in this task and required for optimal task performance, learning this task preferentially involved PPC-A throughout the tone, texture and choice windows, opening additional within-area interaction subspaces, whereas PPC-RL was involved mostly in texture processing. In addition, the interaction subspaces between S1 and PPC-A expanded throughout the task, whereas S1 and PPC-RL interaction was enhanced only during the texture window. These results agree with our previous findings that PPC-A, but not PPC-RL, encodes predictive texture information using the preceding tone, in expert mice performing the same behavioral task²³.” (Line 372-378)

Additionally, while the optogenetic inhibition data suggests both PPC-A and PPC-RL are comparably

important for task performance post-learning, the authors do not present data showing whether these areas contribute during the learning phase itself. Clarifying whether PPC-A plays a more significant role than PPC-RL during learning would strengthen the argument.

Response:

Thank you for this comment. We would like to point out that both PPC-A and PPC-RL were required for optimal task performance, and we do not wish to conclude that PPC-A is more important in task-learning. However, we did observe differences in how PPC-A and PPC-RL activities change during the learning process (see also our answer to the previous point). This does not mean, however, that PPC-A plays a more significant role than PPC-RL during learning, but rather indicates that PPC-A and PPC-RL were involved differentially in improving task performance.

As explained in our response to the previous point, our motivation for emphasizing that PPC-A has a more prominent role in this task originated from our previous study, where we demonstrated that PPC-A, but not PPC-RL, was responsible for tone-texture association in expert mice. In this previous study, we included additional mismatch trials in expert mice, where tone-texture pairing was randomly disrupted in 10-30% of all trials (e.g., tone 1 followed by texture 2, tone 2 followed by texture 1; see Extra Figure 1a-b below, for a copy of Fig. 1b-c in Han and Helmchen, 2024). Using these trials, we were able to deduct the strategy of the mice, with mice choosing either the water port associated with the tone (‘Choose tone’), or the one associated with the texture (‘Choose texture’). In the former case, the texture predictions from the tone was stronger than the actual texture information, whereas in the latter case, the actual texture information was stronger than its predictions (details in Han and Helmchen, 2024).

We have performed additional optogenetic inhibitions in these mismatch trials during expert phase (Extra Figure 1c), and observed that inhibiting PPC-A led to less ‘Choose tone’ trials, whereas inhibiting PPC-RL led to more ‘Choose tone’ trials. This result indicates that PPC-A was responsible for tone-induced texture predictions during the texture window (as inhibiting PPC-A activity in this window led to less prediction-based choices), whereas PPC-RL activity during the texture window provided texture-related information (inhibition of PPC-RL in this window led to more prediction-based choices). These results provide additional evidence that PPC-A and PPC-RL are differentially engaged in performing this task, although both are required for optimal task performance. However, we feel that this optogenetic silencing

Extra Figure 1. (a-b) Replicate of Fig. 1b-c of Han and Helmchen (2024). **(c)** Unpublished results of PPC optogenetic inhibition experiments. Optogenetics inhibition was performed in the texture window (similar to Suppl. Fig. 2 of the revised manuscript). Results are shown for expert mice during mismatched trials, when auditory information and tactile information are in conflict. Inhibition of PPC-A favored behavioral choices based on tactile information, whereas inhibition of PPC-RL favored behavior based on auditory predictions. **(a, b,** and the four task icons in **c** are adapted from Han and Helmchen, “Behavior-relevant top-down cross-modal predictions in mouse neocortex”. Nature Neuroscience, 1-11, 2024, Springer Nature.)

experiment went beyond the scope of this manuscript, and therefore we did not include this result in our manuscript. We can include it if the reviewer feels that it would be appropriate and helpful to clarify the differential roles of PPC-RL and PPC-A. We have not performed any systematic inhibition of PPC areas during training, which would be interesting but which we feel would require a full own study design beyond the scope of this study.

Specific changes in the revised manuscript:

- We have clarified the differences we have observed in PPC-A and PPC-RL, and now emphasize that both areas are required for the task:

“Using simultaneous multi-area two-photon imaging, combined with a sensory discrimination task and longitudinal behavioral and neuronal recordings throughout learning, we provided a systematic overview of learning-related changes in S1 and the higher association areas PPC-A and PPC-RL, both required for optimally performing the task.” (Line 334-337)

“We also observed distinct changes in S1, PPC-RL, and PPC-A that accompanied the improved task representation during learning. S1 and PPC-RL showed decreased neuronal activity dimensionality, decreased within-area axis similarity across time, but stable trial-to-trial variability and increased encoding axis similarity. These findings potentially suggest that the populations compress their variability at each time point to achieve more consistent task encoding, with a dynamic shared variability structure across time. In contrast, PPC-A showed decreased trial-to-trial variability, decreased overall encoding axis similarity, but stable neuronal activity dimensionality and stable within-area axis similarity across time. These adaptations in PPC-A potentially suggest reduced noise levels across trials and diversification of the task-related encoding structure across time, while preserving the variability of the population.” (Line 351-359)

“While both PPC-RL and PPC-A were engaged in this task and required for optimal task performance, learning this task preferentially involved PPC-A throughout the tone, texture and choice windows, opening additional within-area interaction subspaces, whereas PPC-RL was involved mostly in texture processing. In addition, the interaction subspaces between S1 and PPC-A expanded throughout the task, whereas S1 and PPC-RL interaction was enhanced only during the texture window.” (Line 372-376)

Another concern relates to the potential influence of orofacial movements on neural activity and interaction measures. The authors report small but significant changes in facial and licking behavior during learning and across correct vs. incorrect trials. It is crucial to demonstrate that these movements do not confound neural activity or the interaction metrics. Furthermore, do optogenetic inhibitions of PPC-A and PPC-RL impact whisking or licking behavior? Addressing these points would help clarify the interpretation of the neural findings.

Response:

Thank you for this comment. We would like to point out that already in the original manuscript, we had removed all pre-choice early licking frames from our analysis, i.e., if any lick already occurred in the texture window, all time points between the first lick to the onset of the choice window were excluded. This in principle removed the effect of licking behavior during learning. To examine whether this was

also sufficient to account for the differences in facial movement, we have further analyzed the face movements with and without early licking frames. We found that removing early licking frames diminished the differences in facial movement during the tone window. We still observed increased facial movement during the texture window even without early licking frames, however this is likely due to the enhanced and anticipatory whisking that emerges during learning, as routinely observed in previous studies (Chen et al, 2015, Fig. 1g; Gilad and Helmchen, 2020, Fig. 2e-h; Schoenfeld et al, 2024, Supplemental Fig. 2b). We believe that this change in whisking pattern is part of task learning and thus is not in conflict with our neural findings but rather could serve as one mechanism that explains certain neural findings.

Unfortunately, we did not record face movements during optogenetic inhibition. However, we hope the above clarification and extra analysis dissolve the reviewer's concern about the impact of facial and licking behavior.

Specific changes in the revised manuscript:

- We have added a new supplemental figure (Supplemental Fig. 1) showing the facial movement with and without pre-licking frames, and added the corresponding text:

“Removing the data points during early licks (licks before choice window) led to a comparable level of face movement during the tone window across learning phases (Supplemental Fig. 1).”
(Line 102-103)

- We also highlighted in the main text that we removed all early lick frames:

“Since early licks during the texture window were not punished, we removed all neural activity frames from the first early lick until choice window onset in all following analysis, to avoid any confounding influence of licking-related activity.” (Line 142-144)

Finally, the current claim that PPC-A is more critical for tactile discrimination than PPC-RL appears inconsistent with the authors' prior study (Gallero-Salas et al., Neuron 2020). In that work, PPC-A was implicated in auditory discrimination, while PPC-RL was linked to tactile discrimination. While the behavioral paradigms differ (delayed go/no-go vs. 2AFC with no delay), this alone does not fully account for the apparent contradiction. The authors should reconcile these discrepancies.

Response:

We thank the reviewer for pointing out this issue, but we believe the perceived inconsistency is due to a misunderstanding. We do not think that there is any discrepancy in the results of these two studies. The major difference between our study and the Gallero-Salas et al (2020) study is the design of the behavioral task. In our task, each texture was always preceded by a distinct tone (tone 1 followed by texture 1, tone 2 followed by texture 2). Therefore, tone alone was informative about the following texture, allowing mice to develop multisensory tone-texture pairing during learning. In Gallero-Salas et al, each texture was preceded by the same tone, therefore tone alone was not informative about the following texture. We speculate that the multisensory nature of our task contributed to the different engagement of PPC areas, as a growing body of literature already suggests that PPC is a highly flexible area and that the exact task presumably has a strong influence on its engagement.

Specific changes in the revised manuscript:

- We have revised the relevant section in discussion and attempted to clarify these points:

“The higher engagement of PPC-A compared to PPC-RL in our study is possibly due to the task design, where a specific tone was paired with a fixed subsequent texture, allowing mice to form specific tone-texture associations²³. In other behavioral tasks where only one stimulus type is required for the correct decision, PPC-A has been implicated in auditory discrimination, while PPC-RL was linked to tactile discrimination²⁶. However, PPC is a highly flexible area, and changes in the task design could strongly influence whether and how PPC is involved⁴⁶.”

(Line 381-386)

Minor comments:

Several figure panels are difficult to read due to their small size, particularly when trying to identify which comparisons are marked by asterisks. This is especially evident in Figs. 6f-i (right panels) and 7e, g, h (right panels), where the details are challenging to see. Enlarging these panels or providing clearer indications of the statistical comparisons would improve readability.

Response:

Thank you for this comment. We have redesigned the corresponding figures to allow more space for these panels.

Specific changes in the revised manuscript:

- We have redesigned Fig. 5, Fig. 6 and Fig. 7 to allow more space for the small panels.

Reviewer #2 (Remarks to the Author):

This paper studies how neural activity changes in somatosensory area S1 and two areas of posterior parietal cortex (PPC-A and PPC-RL) during auditory and texture learning in mice. The authors study how neural activity within each of the areas, and how the interactions between areas, changes with learning. Furthermore, they study how the strength and axes of task encoding (including tone, texture, choice, reward) in the neural activity change with learning, as well as how these axes align with the within-area and inter-area activity spaces.

This paper brings together state-of-the-art optical imaging methods for recording the activity of many neurons in multiple brain areas simultaneously while mice are engaged in an interesting learning task. Furthermore, they study the activity using population analysis methods (such as PCA and CCA), which recent studies have used to gain deeper insight into the neural activity than what is possible by studying each neuron individually.

As much as I wanted to like this paper, it was a (very) difficult read, for the reasons described below. I felt overwhelmed with the number of comparisons being made, and as a result it was difficult to come away with a clear message of the findings. The authors try to summarize the findings, but the takeaways provided by the authors are too simple for what the data show. This paper reads like a series of exploratory analyses without clear hypotheses or take-away messages that are i) well-reasoned in the context of the brain areas (S1, PPC-A, PPC-RL) and learning task being studied, and ii) supported by the data.

Response:

Thank you for evaluating our manuscript and for the extended comments. We would like to note here that our manuscript was intended to be a descriptive and exploratory study, rather than a hypothesis-driven one. We feel that studies on activity changes during learning, including the changes in interactions between areas, are still scarce. Therefore, exploratory studies are needed as basis of creating and further hypotheses. Our dataset consisted of longitudinal two-photon imaging from two simultaneously recorded neuronal populations, during a multisensory discrimination task, and provided one of the first biological applications of this state-of-the-art imaging approach. This allowed us to study learning-related changes on multiple levels, from single neurons to local populations and to cross-area interactions. In addition, our behavioral task provided extra levels of complexity by comprising both sensory association (tone-texture pairing) and sensorimotor transformation (stimuli to choice sequence) as well as multiple trial types. As one of the first in vivo and longitudinal studies that covered all these scales and behavioral complexity, we wished to provide a panoramic characterization of learning-related changes on these different levels. We believe that this systematic characterization is important by itself. However, we integrated the reviewer's comments and systematically revised our manuscript to provide clearer statement of results as well as interpretations, and to clarify the descriptive and explorative nature of our study.

Specific comments:

1) Throughout the paper, the authors do a large number of statistical tests and just focus on the comparisons that are statistically significant. The authors try to summarize all the comparisons with a sweeping statement, but in each case the statement is far too simple for what the data show. For example, in Fig. 5f, is there a principled reason by which we can understand why PPC-A would significantly increase for only the Texture and Choice curves within the Texture period? What does it mean that the Texture and Reward curves *decrease* during the Tone period? What about all the other comparisons that are not significant - are we allowed to ignore them? How reliable are these effects across individual mice? The same types of questions arise throughout the manuscript, including in Figures 4e, 5c, 6b, 6c, 6f, 6g, 7e, 7g, 7h, Supplemental Fig. 4, and Supplemental Fig. 5.

Response:

Thank you for these comments. In our initial submission, we were limited by the journal guideline that specified a maximum of 5000 words, while we had already reached almost 8000 words. We therefore did not expand the results to discuss all comparisons, but rather to focus on the most relevant observations. Thanks to the reviewer's comment, we have revised our manuscript to include more specific descriptions of the results, as well as their potential implications. However, since this is an observational study that aimed at providing a general picture of learning-related changes, we would like to refrain from making interpretations for every single observation. We tried to strike a balance between the level of detail and the overall picture in our revised manuscript.

To address the reviewer's specific questions:

- In Fig. 5f, we observed an increased alignment between PPC-A texture encoding axis with its within-area interaction axis during the texture and choice window. This indicates that the task-related (texture-related in this case) variability strongly overlapped with the shared population variability, suggesting that the within-area activity in PPC-A during the texture and choice window could carry more texture information that was essential for making the correct choice in this task. This increased alignment was absent during incorrect trials (Fig. 7f), further suggesting that the coordinated task-related activity is associated with correct task performance. In contrast, a decreased alignment could indicate that the task-related variability was less aligned with the shared population variability, which could be a result of orthogonal encoding dimensions to the shared subspace, and/or the shared variability capturing other

processes. We could only speculate that this alignment did not increase in the tone or reward window because the activity in these windows did not contain texture-related variability. We have also observed in other analysis that PPC-A activity during the tone and texture window was reconfigured differently during learning (Fig. 2d-g, Fig.3a-f, Fig. 4d-e, Fig. 5c). We provided some discussions on this topic; however, we believe that it goes beyond the scope of our current manuscript to understand the detailed differences in mechanisms between learning sensory association and learning sensorimotor transformation.

- In Fig. 5f, the alignment decreased between PPC-A tone encoding axis and its within-area axis during the texture and reward windows. This indicates that the tone-related variability was less aligned with the shared population variability during texture and reward windows. This mirrors the changes in the encoding axis structure (Fig. 4d-e), where the tone encoding axis in PPC-A became less similar with texture, choice and reward encoding axes. Again, we could only speculatively interpret that tone representation became more distinct with other task variables, potentially by involving a separate set of neurons. The tone-texture association was not learnt through shared representation, but other mechanisms (for example, priming the interaction subspace between S1 and PPC-A closer to the texture stimulus, as seen in Fig. 6f).
- Regarding all the other comparisons that are not significant, these indicate that there was no significant increase or decrease in the alignment between the task encoding axis and the within-area axis during learning. This suggests that naive and expert conditions did not differ meaningfully in how task-relevant information aligns with within-area shared subspaces. However, this does not exclude learning-related changes on other levels, defined by other measurements.
- For individual variability, we performed analysis on individual mice, and now provide a new supplemental figure to give an overview of individual variability. In this figure, we showed the learning curves for each mouse, together with the discrimination index on the encoding axis, the canonical correlation values, and the significant dimensions for both within-area and inter-area subspaces. We observed variability between individuals, which to a certain degree correlated with the learning speed and the number of sessions recorded. However, pooling results by mouse (averaging all sessions for each mouse and pooling results) did not change the results (see Supplemental Fig. 6, last column). We provided this figure for reference, and included corresponding text in the manuscript.
- For Fig. 4e, we measured the similarity of the encoding axis across trial phases. The encoding axis represents the direction in the neural space that best separates task variables (e.g., stimulus 1 vs 2). Its similarity between pairs of adjacent task windows reflects the consistency of task-related representations over time. An increased similarity reflects shared/stable task representation, as shown between the texture and choice windows. This increase is consistent with the increased percentage of neurons jointly tuned for texture and choice (Supplemental Fig. 3). A decreased similarity reflects dynamic/evolving task representations over time, as shown between tone window and the following task windows in PPC-A.
- For Fig. 5c and Fig. 6b, we measured the canonical correlation value on the top CCA axis. This represent the coordination between the two populations, either within the same area (Fig. 5c) or between S1 and PPC (Fig. 6b). An increased correlation value represents more aligned and coordinated neural activity, suggesting more consistent encoding by the two populations; an decreased correlation represents less coordinated activity, potentially due to private variability caused by noise or independent representations.
- For Fig. 6c, we measured the number of significant CCA dimensions for S1-PPC interactions (similar as defined in Semedo et al, 2019), representing the shared information space between the two areas during

the task. An increased dimensionality suggests increased coordination and communication between the two populations, potentially due to that the two populations were both engaged in common computational process driven by the common task inputs. This could also suggest more complex but coordinated task activity in both populations.

- For Fig. 6e and Fig. 6f (previously Fig. 6f and Fig. 6g), using a similar measurement as Fig. 5f, we observed increased alignment between PPC-RL texture/choice encoding axes and its S1-interaction subspace during the texture, choice and reward windows, as well as between PPC-A texture/choice encoding axes with its S1-interaction subspace during tone, texture, and choice. This suggests that the task-related variability in PPC aligned with its co-variability with S1, potentially leading to more efficient and coordinated information transfer between these two areas. In PPC-A, this increased alignment was observed during the tone window, whereas in PPC-RL it occurred only after the texture window. This hints towards the role of PPC-A in priming its activity during the tone window to align with the representations for later task variables, which could serve as evidence of stronger temporal continuity during trials and predictive processing (as we observed in a previous study). This increased alignment was absent during incorrect trials (Fig. 7g-h), suggesting that the efficient and coordinated information transfer is associated with correct task performance.
- Supplemental Fig. 7 (previous Supplemental Fig. 4) is discussed extensively in our response to the next point.
- In Supplemental Fig. 8 (previous Supplemental Fig. 5), we observed similar pupil diameter between correct and incorrect trials in expert condition, but slightly increased face movement during tone and reduced face movement during choice. We now included this more precise summary in our manuscript.

We hope that these clarifications explain our view of the results, and that the above discussions could dissolve the reviewer's concerns.

Specific changes in the revised manuscript:

- We have revised the results section to include specific descriptions of the results and to clarify the comparisons being made, as well as the possible implications and our interpretations. We added as much as possible of this information without hindering the overall flow of the manuscript. These revisions include:

“Furthermore, the encoding axes from adjacent task windows became more aligned over learning across S1 and PPC areas (Fig. 4d-e), suggesting shared task representation between adjacent windows.” (Line 216-217)

“In PPC-A, the encoding subspace from non-adjacent task windows even became more distinct from each other, suggesting more distinct representations in non-adjacent windows.” (Line 219-220)

“Compared to naive sessions, the within-area interaction strength slightly decreased in S1 in expert sessions, but increased in PPC-RL during the texture window, and increased in PPC-A during both texture and choice windows (Fig. 5c). These results suggest stronger shared activity within PPC, but diverse subgroups of activity patterns in S1.” (Line 233-236)

“The identified significant interaction dimensions captured the instantaneous co-activity between populations.” (Line 242-243)

“With learning, the within-area subspaces in S1 and PPC-A significantly increased their dimensionality across all task windows, suggesting more coordinated neural activity and consistent encoding patterns, while PPC-RL showed an increase only during the texture window (Fig. 5d). This is consistent with our previous finding that PPC-A but not PPC-RL forms tone-texture associations and generates sensory predictions²³.” (Line 244-247)

“we computed the cosine similarity between the encoding axis and the top within-area axis computed from the same subpopulation (Fig. 5e), reflecting the alignment between the task-related variability and the within-area population variability.” (Line 250-253)

“In PPC areas, the texture encoding axis became more aligned with the within-area axes during the texture and choice windows (Fig. 5f), indicating that the within-area activity in PPC during these task windows could carry more texture information, which is essential for the correct choice. In contrast, the tone encoding axis in PPC-A became more distinct from the within-area axes in the following task windows, coinciding with the changes in encoding axis structure.” (Line 255-259)

“Over the course of learning, the interaction strength between S1 and PPC-RL slightly decreased during the tone and reward windows, while their interaction subspace dimension expanded during the texture window (Fig. 6b).” (Line 278-280)

“Additionally, the interaction subspace dimensionality increased for S1 and PPC-A during the tone, texture and choice windows (Fig. 6c), suggesting enhanced coordination and communication, potentially due to the engagement of both populations in a common computational process driven by the common task inputs. This enhanced communication of S1 and PPC-A is also consistent with our previous finding of strong interactions between S1 and PPC-A, but not PPC-RL, during sensory predictions²³.” (Line 282-286)

“In PPC-RL, its inter-area axes with S1 after the texture onset became more aligned with the texture and choice encoding axes (Fig. 6e), suggesting more efficient and coordinated information transfer between S1 and PPC-RL. In PPC-A, this improvement in alignment occurred earlier during the trials, prior to texture onset, where the inter-area axes during tone window developed texture and choice information over learning (Fig. 6f). Such improvement also occurred in S1 interaction subspace with PPC-A. These results suggest that PPC-A forms sensory associations between tone and texture, and plays an important role in multisensory processing in this specific task.” (Line 289-295)

“In these expert incorrect trials, the licking pattern as well as the pupil diameter and face movement of mice were different from the incorrect trials in naive sessions, but were overall similar with the expert correct trials, despite slightly increased face movement during tone and reduced face movement during choice (Supplemental Fig. 8).” (Line 309-312)

“these axes were less aligned with the task-encoding axes compared to correct trials in S1 and PPC-A (Fig. 7d-h), both in their within-area interaction subspace (Fig. 7f) and in their communication subspace (Fig. 7h), while PPC-RL was not affected (Fig. 7f,g).” (Line 325-327)

- We emphasized the explorative and descriptive nature of this manuscript:

“To provide an overview of learning-related changes from this rich dataset, we systematically examined four types of population subspaces” (Line 65-67)

“Using simultaneous multi-area two-photon imaging, combined with a sensory discrimination task and longitudinal behavioral and neuronal recordings throughout learning, we provided a systematic overview of learning-related changes in S1 and the higher association areas PPC-A and PPC-RL, both required for optimally performing the task. We discovered changes on different levels, supporting multiple learning mechanisms.” (Line 334-338)

- We added a new supplemental figure (Supplemental Fig. 6) to show the results from individual mice, as well as results pooled by mice instead of sessions. The information was also added to the main text:

“Individual mice showed variability in these observations that corresponded to their learning rate, however pooling the results by averaging each mouse did not change our observations (Supplemental Fig. 6)” (Line 214-216)

2) The axis similarity metric, which is used throughout in the paper (e.g., Fig. 4e, Supplementary Figure 4), is difficult to interpret. If this metric changes from novice to expert, it could either be that one axis changes or both axes change. Conversely, if this metric is unchanged from novice to expert, it still could be that both axes change (but the similarity between them remains unchanged). For this reason, the following key claims are not supported by the current analyses:

L247: "We found that ...remained stable in PPC-A"

L249-251: "these results suggest that...during learning"

L276: "mostly remained stable over the learning process"

Response:

We thank the reviewer for pointing out this issue. We fully agree with this point. We had considered the alternative explanation but did not use appropriate writing in interpreting the results. The appropriate conclusion, as pointed out by the reviewer, is that the similarity structure of the interaction axes remains unchanged. We would also not consider the above claims to be key claims, as our main results were based on comparing the encoding axes with the interaction axes, whereas these supplemental results provided complimentary information on the pairwise similarity of these axes alone.

While we observed increased alignment (similarity) between task encoding axes and within-/inter-area axes during learning, we indeed could not disentangle whether this was due to a change in one of the axes or both axes. In fact, we reported some changes in both: on one hand, we observed improved discrimination index on the encoding axis as well as altered encoding axis similarity structure; on the other hand, we observed expanded interaction subspaces, but similar or reduced correlation strength as well as similar or decreased interaction axis similarity structure. We have now modified the results section to accurately reflect the changes we observed. We hope these clarifications and improved writing dissolve the reviewer's concerns.

Specific changes in the revised manuscript:

- We revised the results section and summarized the findings in Fig. 4e and Supplemental Fig. 7 (previously Supplemental Fig. 4) with unchanged axis similarity structure, instead of unchanged axis:

“We found that the within-area axis structure across task phases did not show increased alignment in PPC-A, and even became more distinct in S1 and PPC-RL over the learning process

(Supplemental Fig. 7a), indicating distinct neuronal patterns across task windows in S1 and PPC-RL. Together, these results suggested an improved alignment between task encoding subspaces and within-area subspaces in PPC during learning, presumably driven by an optimized encoding subspace structure (Fig. 4d-e) and potentially contributing to improved task representation by the local populations.” (Line 261-266; previously line 247-251)

“We found that the alignment between inter-area interaction axes across task windows did not change significantly overall during learning (Supplemental Fig. 7b-c).” (Line 297-299; previously line 276)

3) The authors need to be clearer about how they ruled out alternative explanations of their results. For example:

a) In several analyses, the authors perform a shuffle analysis as a control. In each case, they should carefully explain what the shuffle is designed to do, and justify why it is the right control for that analysis.

Response:

We thank the reviewer for this comment. We performed shuffle controls in the following analysis: (1) finding task responsive neurons; (2) computing discrimination index on the encoding axis; (3) computing CCA models for within- and inter-area interaction analysis. The purposes of these shuffled controls are as following:

(1) To find task responsive neurons, we compared the activity of each neuron during task window T to its own shuffled baseline. To preserve the firing pattern of each neuron, we generated repeated shuffled controls by randomly sampling the same amount of frames as in task window T, but from other task windows. We identified task responsive neurons as those with higher firing rate in T than the 95% quantile of the shuffled distribution from outside of T.

(2) To determine whether the discrimination index was above chance level, we permuted the neuron identity of each trial while preserving the task structure, and computed the discrimination index using the original encoding axes. These shuffled controls preserved the population firing rate during the task and served as a null distribution for the discrimination index.

(3) For CCA controls, we aimed to capture subspaces that can be maximally explained by task-dependent changes in neuronal covariance, but not instantaneous co-fluctuations. To generate such controls, we randomized the trial correspondence between the two populations, while preserving the temporal structure of the neuronal activity during the task. These shuffled models captured the interaction subspaces resulting from the change in activity level due to task factors, e.g., sensory inputs, choice and reward, but not from the simultaneous interaction between populations.

Specific changes in the revised manuscript:

- We added the above information in the results and methods sections:

“The shuffled distribution was generated by taking the top canonical correlation from shuffled models, where the trial correspondence was randomized but the overall population firing rate during the task was preserved.” (Line 240-242)

“To generate a null distribution of the discrimination index, we shuffled the neuronal ordering of the population activity, and computed the discrimination index using the encoding axis from the

real data. This shuffling procedure disrupted the contribution of individual neurons to task encoding, while preserving the overall firing rate of the population during the task.” (Line 687-690)

“To identify interaction subspaces that reflect the instantaneous co-fluctuation between two populations, we computed 100 shuffled models by randomizing the trial correspondence between the two populations, while preserving the temporal structure of neuronal activity during the task. These shuffled models captured the interaction subspaces resulting from the change in activity levels due to task factors (sensory inputs, choice, reward), but not from the simultaneous interaction between populations. The highest correlation values from these shuffled models represent the shared activity that can be explained by the task structure.” (Line 708-714)

b) Can some of the results (e.g., Fig 4d,e, Fig. 5f, Fig. 6) be solely explained by increases in firing rates during learning?

Response:

In Fig. 4d-e, Fig. 5f and Fig. 6, we computed the similarity between task encoding axes (Fig. 4), between within-area axes and task encoding axes (Fig. 5), and between inter-area axes and task encoding axes (Fig. 6). These three types of axes were all computed using z-scored neural activity. This normalization procedure should, in principle, remove the influence of different levels of firing rate. As the reviewer suggested in a later point, to capture the influence of global changes in firing rate, we also computed the similarity between these three types of axes and the [1, 1, 1, ...] uniform axis. We did not find a significant difference between the naive and expert conditions (see Extra Figure 2 below). However, as we believe that the z-score procedure already removed the influence of firing rate changes in the original manuscript, we did not include this figure in the revised manuscript. We can include it if the reviewer feels that it would be appropriate and helpful to further clarify the contribution of firing rate changes.

Extra Figure 2. Axis similarity with the [1, 1, 1, ...] axis. (a) Encoding axis similarity with the [1, 1, 1, ...] axis across task windows, in naive (gray) and expert (colored) conditions. Encoding axis was compared to the constant axis to capture the influence firing rate changes in measured similarity values. (b) Within-area axis similarity with the constant axis. (c) Inter-area axis similarity with the constant axis. (S1: 13 mice, 122, 119 sessions [naive, expert]; PPC-RL: 13 mice, 66, 86 sessions; PPC-A: 10 mice, 56, 33 sessions; S1 and PPC-RL interaction: 13 mice, 66, 86 sessions; S1 and PPC-A interaction: 10 mice, 56, 33 sessions; mean±SEM; *p<0.05; n.s. p>0.05; Wilcoxon rank-sum test against naive condition).

Specific changes in the revised manuscript:

- We clarified the z-score procedure in the results section:

“We extracted $\Delta F/F$ traces as well as the deconvolved spike rates of individual neurons using Suite2p³¹, and all following analysis concerning neuronal activity was performed on the z-scored spike rates. This z-score procedure, performed independently for each session, in principle removed the effect of changes in population firing rates across learning for all following analysis.”
(Line 112-115)

c) Do the results in Fig. 5f follow trivially from the results in Fig. 3g because the identified within-area axis structure is sensitive to task encoding?

Response:

In Fig. 3g, we showed that the top PCA components increased their task variable encoding, as measured by the discrimination index of task variables. In Fig. 5f, we showed that the within-area interaction axis, defined as the top CCA axis of two random subpopulations within an imaged area, increased its alignment with the task encoding axis. These two observations are linked, as the PCA results suggested an increased variance in the task-relevant direction, while the CCA results suggested an increased shared task-relevant variability across subpopulations. However, we do not view these similar results as trivial, since PCA captures the structure of total variability in the population, while CCA captures the consistent and reproducible shared variability between subpopulations, especially using the averaged results from multiple random splits. We rather view these similar results from two different methods as mutually supporting, and we believe these observations should strengthen our conclusion that the population variability aligned more with task-relevant direction.

Specific changes in the revised manuscript:

- We expanded the methods section to include a discussion comparing PCA and CCA methods:

“It is worth noting that PCA of the neuron space also captures features of within-area interactions, similar to CCA performed on two subpopulations from an area. However, these two methods reflect different characteristics of the data, with PCA dimensionality capturing the total variability structure during the task, including both shared and private variability, while CCA dimensionality capturing the reliable shared variability between subpopulations. When the task-relevant variability dominates the total variance and is shared consistently across neurons, both methods define similar axes.”
(Line 724-729)

d) Are the number of neurons being analyzed matched over time (novice vs. expert) within each brain area? How do differences in the number of neurons recorded in each area influence the results?

Response:

We thank the reviewer for this important comment. We indeed observed a decrease in imaged neuron numbers during learning, possible due to the decay in cranial window quality over time (see Extra Figure 3 below).

This change in neuron numbers could potentially influence the PCA dimensionality analysis in Fig. 3. To address this issue, we re-analyzed the population dimensionality by matching the number of naive and expert sessions. For each naive session, we randomly resampled the neuronal population size to match one of the expert sessions, and performed the analysis in Fig. 3. We repeated this resample procedure 50 times and averaged the results from all repeats. The number of neurons resulting from the resample procedure were not significantly different between naive and expert conditions. With this new analysis, we observed a decreased neuronal dimensionality in S1 and PPC-RL during learning, but not in PPC-A. These results suggested neuronal correlation increased in S1 and PPC-RL, but PPC-A activity patterns remained diverse. For trial dimensionality, we observed a decrease in PPC-A but not in S1 and PPC-RL. This suggested that trial-to-trial variability decreased in PPC-A, but not in S1 and PPC-RL. We updated Fig. 3 and the results section accordingly.

For the analysis of encoding axis and interaction axis, we performed all analysis using the same number of variables from each area (top 30 PCs). This ensured matching number of variables across areas and sessions, therefore the change in total recorded neuronal number should not alter the results. We also note that taking a different number of PCs (60 instead of 30) did not change our main results. For a discussion regarding the number of PCs, please see our answer to a later point.

Extra Figure 3. Number of imaged neurons in naive and expert mice. Boxplots show the 25% and 75% quantile of distribution, as well as median; whiskers indicate the extremes of the distribution. Gray boxes represent naive sessions; colored boxes represent expert sessions. (S1: 13 mice, 122, 119 sessions [naive, expert]; PPC-RL: 13 mice, 66, 86 sessions; PPC-A: 10 mice, 56, 33 sessions; *** $p < 0.001$; Wilcoxon rank-sum test between naive and expert conditions).

Specific changes in the revised manuscript:

- We updated Fig. 3 and the corresponding result section and discussion:

“To account for the reduced number of recorded neurons in the expert phase (due to the decay in imaging quality over longitudinal imaging), we resampled the neuronal populations in naive and learning sessions to match one of the expert sessions, and repeated this procedure 50 times.”

(Line 168-171)

“Compared to naive sessions, the population dimensionality in expert sessions consistently decreased in S1 and PPC-RL areas in all task windows, but remained unchanged in PPC-A (Fig. 3b-c). This result suggests that in expert sessions, neurons in S1 and PPC-RL developed correlated firing patterns in each task window, while the activity patterns in PPC-A remained diverse. We further analyzed the dimensionality in the trial space by performing PCA on the population activity

with trials as variables and neurons as observations (Fig. 3d). This analysis directly measures the neuronal reliability across trials. While the dimensionality remained unchanged in S1 and PPC-RL during the learning process, it decreased in PPC-A (Fig. 3e-f), indicating decreased trial-to-trial response variability.” (Line 174-181)

“On the population level, neuronal response dimensionality and trial-to-trial variability decreased differentially in S1 and PPC.” (Line 339-340)

4) Similar within-area questions are asked using different methods in Fig. 3 (PCA) and Fig. 5 (CCA, by arbitrarily dividing the population into two subsets), making the resulting analyses conceptually redundant. How does one reconcile the sometimes different results obtained by the two methods? For example:

- reconciling Fig. 3c (middle) with Fig. 5d (middle), which look inconsistent with each other.

Response:

In Fig. 3c, we performed PCA on the population activity and quantified the percentage of PCs required to explain 70% total variance. We observed in the updated Fig. 3c (middle) that PPC-RL required less components to explain 70% variance during learning, indicating that PPC-RL neurons developed correlated firing patterns. In Fig. 5d, we quantified the number of significant within-area CCA dimensions during learning. We note here that this analysis captures significant shared variability in the population, where PCA captures both independent and shared variability across neurons. We observed an increased CCA dimensions, indicating consistent encoding across subpopulations. These results are in agreement with each other, as they both indicate that the population became more correlated, and the task information was encoded more efficiently by subpopulations.

Specific changes in the revised manuscript:

- Fig. 3 has been updated (see details above, in Point 3d) showing these consistent results.

- relating Fig. 3g to Fig. 5h.

Response:

In Fig. 3g, we quantified the task discrimination index of top PCs, and observed an increased task information in top PCs during learning. In Fig. 5h, we quantified the task discrimination index of population activity projected onto the within-area axis, and observed an increase during learning as well. Following our response to Point 3c above, these results are related and consistent with each other. Conceptually, PCA captures the total variability in the population including both shared and independent variability, while CCA performed on subpopulations captures the shared variability. In an ideal scenario, where the task-relevant variability dominates the total variance and is shared consistently across neurons, both methods should reflect a similar trend. Because of the different nature of these two methods, we still applied both in our analysis, and we believe that consistent results from both methods should strengthen our findings.

Specific changes in the revised manuscript:

- We included a discussion regarding the similarities and differences between PCA and CCA methods, in the methods section:

“It is worth noting that PCA of the neuron space also captures features of within-area interactions, similar to CCA performed on two subpopulations from an area. However, these two methods reflect different characteristics of the data, with PCA dimensionality capturing the total variability structure during the task, including both shared and private variability, while CCA dimensionality capturing the reliable shared variability between subpopulations. When the task-relevant variability dominates the total variance and is shared consistently across neurons, both methods define similar axes.” (Line 724-729)

5) I tripped in a large number of locations, which made this paper difficult read. Here are some examples of where I tripped:

L130-131: the logic is unclear why this suggests "both areas are involved in tone-texture association".

Response:

We thank the reviewer for the careful reading and thoughtful notes. We apologize for the confusion regarding these ‘tripping’ points, many of which were caused by our attempt to compress the writing in the results section due to space limitation, while still trying to emphasize the most noticeable findings (in our mind). We have systematically revised the results section to be more precise on the results.

For this first question, inhibition of PPC-RL and PPC-A both reduced performance when only the tone was presented, without the presence of the texture. In our previous study using the same cohort of mice, we have shown that these mice learnt tone-texture association at the end of expert phase, and that tone alone could elicit texture representation in PPC-A (Han and Helmchen, 2024). Due to these previous findings, we used the phrase “tone-texture association” in our interpretation. We have modified our writing here, and generally included more information on our previous study to provide sufficient background and justifying our interpretations.

Specific changes in the revised manuscript:

- We revised the above-mentioned writing and cited our previous study:

“Inhibition of either PPC-RL or PPC-A reduced mouse performance in tone-only condition where no texture was presented (Supplemental Fig. 2c), suggesting that both areas were involved in transforming sensory association to decision²³.” (Line 127-129)

L151-152: "PPC-A developed a higher percentage..." How is this statement supported by what is shown in Fig. 2d?

Response:

Thank you for this comment. We apologize for the imprecise summary, and we have revised the text to accurately reflect the results.

Specific changes in the revised manuscript:

- We revised the writing to describe the results in more specific details:

“While PPC-RL showed an increased percentage of texture-responsive neurons, PPC-A developed a higher percentage of tone-, texture- and choice-responsive neurons, indicating its role as a higher-order area in this task.” (Line 149-151)

L153: "showed a similar trend" There are so many differences between Fig. 2e and Fig. 2d that are being ignored by the authors.

Response:

Thank you for this comment. We have revised the text to describe the results in more details.

Specific changes in the revised manuscript:

- We revised the writing to describe the results in more specific details:

“Task-discriminative neurons in these areas also increased during learning, with PPC-A overall containing the highest percentage of task-discriminative neurons for tone, choice and reward (Fig. 2e).” (Line 151-153)

Fig 2f: is the definition of each type of neuron based on the naive or expert period? Is the definition based on if the neuron is task-responsive or task-discriminative?

Response:

Due to experimental constraints, we recorded slightly different neuronal populations each day. These neurons were recorded from a similar field-of-view for each mouse (Fig. 2b), but we were unable to match the neurons throughout the long learning curve. Therefore, the definition of each neuron’s type was made separately in each session. The traces shown in Fig. 2f were using task-discriminative neurons.

Specific changes in the revised manuscript:

- We clarified the session-based analysis in the methods section:

“This procedure was performed independently for each session.” (Line 652-653)

- The figure legend of Fig. 2f specifies that these traces were from task-discriminative neurons:

“(f) Session-average spike rates of task-discriminative neurons over trial time.”

Fig 3b: Is the vertical axis the number of dimensions or a percentage?

Response:

The axis in Fig. 3b represents the percentage of PCs required to explain 70% of total variance. This percentage was computed by finding the number of PCs that explains 70% variance, divided by the total number of variables.

Specific changes in the revised manuscript:

- The figure legend of Fig. 3b specifies the percentage as a measurement:

“(b) Percentage of principle components (PCs) that explains 70% of variance in neuron subspace”

Fig 3d-f: what is the logic of analyzing activity in this trial space (where each dot is neuron)? If the goal is to assess trial-to-trial variability, why not analyze it in the neuron space, as is standard in our field?

Response:

For PCA in the trial space, each neuron is a data point and trials are variables. In this configuration, PCA captures variability of how individual neurons respond across trials and reveals the trial-to-trial variability. This analysis directly measures the neuronal reliability across trials. For PCA in the neuron space, each trial is a data point and neurons are variables, and PCA captures the population-wide shared variability. This analysis identifies coordinated changes across neurons and the dominant patterns in the activity. We believe that both analysis approaches are valid and reveal different characteristics of the data.

Specific changes in the revised manuscript:

- We revised the corresponding text to integrate more details and reasoning, and updated the schematics in Fig. 4a and Fig. 4d to reflect more details:

“For the dimensionality of the neuron space, we performed principal component analysis (PCA) on the population activity, with neurons as variables and trials as observations at each time point of the trial, for each session separately (Fig. 3a). This analysis identifies coordinated changes across neurons and the dominant patterns in the activity.” (Line 171-174)

“We further analyzed the dimensionality in the trial space by performing PCA on the population activity with trials as variables and neurons as observations (Fig. 3d). This analysis directly measures the neuronal reliability across trials.” (Line 177-179)

L184: "...through redundant representation" - I don't understand this logic.

Response:

Thank you for this comment. We meant high-dimensional representation, where multiple channels are used to represent task information. We have revised the text accordingly.

Specific changes in the revised manuscript:

- We revised the corresponding text:

“suggesting improved task information encoding through high-dimensional representation” (Line 186-187)

L213: should this read "LESS distinct from each other"?

Response:

The corresponding Fig. 4d-e showed that the similarity decreased in non-adjacent windows in PPC-A, indicating different encoding axes in these non-adjacent windows. We revised the corresponding writings to more accurately describe these results.

Specific changes in the revised manuscript:

- We revised the corresponding text:

“In PPC-A, the similarity between encoding axes from non-adjacent task windows decreased, suggesting more distinct representations in non-adjacent windows.” (Line 219-220)

L221: CCA finds uncorrelated dimensions, not orthogonal dimensions.

Response:

Thank you for this comment. We have corrected the corresponding text.

Specific changes in the revised manuscript:

- We revised the corresponding text:

“CCA finds uncorrelated sets of projection axes within each population” (Line 228-229)

L227-229: how to interpret these observations?

Response:

The canonical correlation value reflects the correlation between the activity of two populations. Higher correlation values indicate that the two populations reliability share structured variability. We have modified the corresponding writings and added our interpretations for the results.

Specific changes in the revised manuscript:

- We revised the corresponding text:

“Compared to naive sessions, the within-area interaction strength slightly decreased in SI in expert sessions, but increased in PPC-RL during the texture window, and increased in PPC-A during both texture and choice windows (Fig. 5c). These results suggest stronger shared activity within PPC, but diverse subgroups of activity patterns in SI.” (Line 233-236)

L232: why take only the top canonical dimension?

Response:

This question concerns the generation of shuffled CCA models, where we permuted the trial correspondence of the two populations, while preserving the task structure. The top canonical correlation from these shuffled models reflects the maximum variance that can be explained by the task structure alone, without the co-fluctuation of the two populations. This serves as a control to identify the interaction between populations that are not purely a result of the common task structure.

Specific changes in the revised manuscript:

- We revised the corresponding text and methods to include more explanations:

“The shuffled distribution was generated by taking the top canonical correlation from shuffled models, where the trial correspondence was randomized but the overall population firing rate during the task was preserved. The identified significant interaction dimensions captured the instantaneous co-activity between populations.” (Line 240-243)

“These shuffled models captured the interaction subspaces resulting from the change in activity levels due to task factors (sensory inputs, choice, reward), but not from the simultaneous interaction between populations. The highest correlation values from these shuffled models represent the shared activity that can be explained by the task structure.” (Line 711-714)

L234: why does higher-order area imply higher dimensionality? this is not obvious.

Response:

We apologize for the imprecise writing. We have removed this claim from the text.

Specific changes in the revised manuscript:

- We removed this claim from the text.

L237: "expanded and enhanced" - what does this mean?

Response:

We apologize for the imprecise writing. We wished to state that task-learning expanded within-area interaction subspace and strengthened within-area interactions. We have revised the text accordingly.

Specific changes in the revised manuscript:

- We revised the corresponding text:

“We conclude that task-learning expanded the within-area interaction subspace and strengthened the within-area interactions in PPC.” (Line 247-248)

L247: "became more distinct" - what does this mean, and how should the reader interpret these changes?

Response:

We computed the similarity between within-area axes from different task windows, which reflects the similarity of dominant activity structure across task windows. A decreased similarity indicates more distinct activity patterns from the corresponding task windows, suggesting that population activity changed dynamically during different task phases. We have revised the text accordingly to include our interpretations.

Specific changes in the revised manuscript:

- We revised the corresponding text:

“We also quantified the changes in the within-area subspace structure by computing the similarity between pairs of within-area axes, which reflects the similarity of shared activity patterns across task windows.” (Line 259-261)

“We found that the within-area axis structure across task phases did not show increased alignment in PPC-A, and even became more distinct in S1 and PPC-RL over the learning process

(Supplemental Fig. 7a), indicating distinct neuronal patterns across task windows in S1 and PPC-RL.” (Line 261-264)

L261-267: are these results what the authors would have predicted? how do we understand these results? I'm having a hard time wrapping my mind around what's increasing, what's decreasing, and what's staying the same.

Response:

Here we reported a slightly decreased correlation strength during tone and reward in PPC-RL, but otherwise unchanged values in other task windows and in PPC-A. We also reported expanded interaction subspace dimensions during the texture window for S1 and PPC-RL interactions, and during the tone, texture, and choice windows for S1 and PPC-A interactions. In our previous study (Han and Helmchen, 2024), we reported that PPC-A was involved in forming tone-texture association and was generally more involved in encoding tone information. Based on these previous findings, our observation here would agree with the role of PPC-A, but not PPC-RL, in encoding tone and potentially generating tone-texture association.

We realize that we had to make many comparisons to accurately describe the results. As an explorative and descriptive study, we try to find a balance between covering the details and still giving an overview of the process. We therefore added some discussions to summarize the main changes we observed in S1 and PPC areas during learning.

Specific changes in the revised manuscript:

- We revised the corresponding result section to provide our interpretations:

“Over the course of learning, the interaction strength between S1 and PPC-RL slightly decreased during the tone and reward windows, while their interaction subspace dimension expanded during the texture window (Fig. 6b). In contrast, S1 and PPC-A interaction strength remained stable and consistently higher compared to S1 and PPC-RL interactions. Additionally, the interaction subspace dimensionality increased for S1 and PPC-A during the tone, texture and choice windows (Fig. 6c), suggesting enhanced coordination and communication, potentially due to the engagement of both populations in a common computational process driven by the common task inputs. This enhanced communication of S1 and PPC-A is also consistent with our previous finding of strong interactions between S1 and PPC-A, but not PPC-RL, during sensory predictions²³.”
(Line 278-286)

- We also added a paragraph in discussion to summarize the learning-related changes in each area, focusing on their differences:

“We also observed distinct changes in S1, PPC-RL, and PPC-A that accompanied the improved task representation during learning. S1 and PPC-RL showed decreased neuronal activity dimensionality, decreased within-area axis similarity across time, but stable trial-to-trial variability and increased encoding axis similarity. These findings potentially suggest that the populations compress their variability at each time point to achieve more consistent task encoding, with a dynamic shared variability structure across time. In contrast, PPC-A showed decreased trial-to-trial variability, decreased overall encoding axis similarity, but stable neuronal activity dimensionality and stable within-area axis similarity across time. These adaptations in PPC-A

potentially suggest reduced noise levels across trials and diversification of the task-related encoding structure across time, while preserving the variability of the population.” (Line 351-359)

Fig 6c: for dimensionalities less than 1, does it mean that the authors are consistently finding that there are no across-area dimensions above chance? How are the subsequent analyses performed in this case?

Response:

The dimensionalities are averages of 20 repeats. An average dimensionality less than 1 does not mean that none of the repeats finds significant dimensions. In cases where no significant interaction dimensions were found, the top CCA axes still represent the optimal interaction subspaces, although these subspaces do not contain more information than the effect of common task inputs. This is also particularly meaningful in the naive condition. As we performed the subsequent analysis using each repeat independently, we averaged all results even if the interaction dimension was not significant.

Specific changes in the revised manuscript:

- We added this information in the methods section:

“In cases where no significant interaction dimensions were found, the top CCA axes still represent the optimal interaction subspaces, although these subspaces do not contain more information than the effect of common task inputs. Subsequent analysis was performed by always taking the top axes.” (Line 716-718)

Other questions:

L200: what is the percentage of variance retained by the top 30 PCs? Is this preprocessing step capturing almost all of the variance in the population activity, as this could influence the CCA dimensions found?

Response:

We quantified the variance retained by the top 30 PCs as following: in S1, the variance is 41.4±2.0 (naive, mean±SEM, in %), 41.7±1.5 (learning), 51.7±2.2 (expert); in PPC-RL, 37.0±2.6 (naive), 38.5±2.6 (learning), 47.1±2.5 (expert); in PPC-A, 38.0±1.6 (naive), 42.7±1.4 (learning), 50.8±2.9 (expert).

The amount of variance retained could potentially restrain the initial space of CCA analysis and affect the CCA dimensions if too much shared variance were excluded. As a control, we also performed additional analysis using the top 60 PCs, where the retained variance was 63.6±2.0, 65.6±1.7, 75.1±2.1 in S1, 58.5±2.9, 60.3±2.7, 72.0±2.5 in PPC-RL, 62.2±2.2, 68.3±1.7, 77.6 ±/− 2.8 in PPC-A (naive, learning, expert, correspondingly; mean±SEM, in %). We were able to reproduce the main findings in the manuscript using top 60 PCs.

During learning, the top 30 PCs (or 60 PCs) showed increasing amounts of retained variance. However, we tend to view this increased variance in top PCs as one of the potential mechanisms underlying the learning process, rather than a confounding factor. As we analyzed the learning process on multiple scales and levels, we believe that the learning-related changes were linked to each other, contributing to the common result of increased task performance.

Specific changes in the revised manuscript:

- We added a new Supplemental Fig. 4 showing the variance explained by the top 30 and 60 PCs, with the main results using top 60 PCs instead, and included a corresponding text:

“Using more PCs did not change the main findings in the following sections (Supplemental Fig. 4).” (Line 204-205)

- We also added the retained variance in the methods section:

“The top 30 PCs explained total variance of 41.4 ± 2.0 (naive, mean \pm SEM, in %), 41.7 ± 1.5 (learning), 51.7 ± 2.2 (expert, same order for the following) in S1, 37.0 ± 2.6 , 38.5 ± 2.6 , 47.1 ± 2.5 in PPC-RL, and 38.0 ± 1.6 , 42.7 ± 1.4 , 50.8 ± 2.9 in PPC-A. Taking top 60 PCs did not change our main results (Supplemental Fig. 4; explained variance: 63.6 ± 2.0 , 65.6 ± 1.7 , 75.1 ± 2.1 in S1, 58.5 ± 2.9 , 60.3 ± 2.7 , 72.0 ± 2.5 in PPC-RL, 62.2 ± 2.2 , 68.3 ± 1.7 , 77.6 ± 2.8 in PPC-A).” (Line 676-680)

How similar are each of the axes (encoding axes, within-area axes, inter-area axes) to the [1,1,1...] direction? This would inform whether the identified axes are driven by general increases in firing rates across the population.

Response:

Thank you for this suggestion. We analyzed the similarity between the [1, 1, 1, ...] axis with the encoding axes, within-area axes, and inter-area axes over the learning process. We did not observe a significant change in the similarity, suggesting that the identified axes were not driven by general increases in firing rates across the population. The results are shown above in Extra Figure 2. However, as we believe that the z-score procedure in our original manuscript already removed the influence of firing rate changes, we did not include this figure in the revised manuscript. We can include it if the reviewer feels that it would be appropriate and helpful to further clarify the contribution of firing rate changes.

Specific changes in the revised manuscript:

- We clarified the z-score procedure in the results section:

“We extracted $\Delta F/F$ traces as well as the deconvolved spike rates of individual neurons using Suite2p³¹, and all following analysis concerning neuronal activity was performed on the z-scored spike rates. This z-score procedure, performed independently for each session, in principle removed the effect of changes in population firing rates across learning for all following analysis.” (Line 112-115)

Reviewer #3 (Remarks to the Author):

The authors investigate the neuronal population dynamics that accompany and potentially support learning of a sensory discrimination task. They find that task learning is marked by an increased number and stabilized responses of individual task neurons as well as increased dimensionality and reduced trial-to-trial variability of population responses. Further, task learning is accompanied by changes of within-area as well as between-area subspaces. Most markedly, the authors demonstrate that task encoding and cross-area interaction subspaces are related. Finally, behavioral errors go along with decreased neuronal responses, decreased encoding accuracy and misaligned subspaces.

As far as I know the literature, I think the present work adds substantial evidence to the growing body of knowledge on how neuronal populations support cognition in general and during learning in particular. I find the multi-level description presented ranging from encoding properties of single neurons and local populations to cross-area interactions between populations to be encompassing and illustrative. Last but not least, the investigation of how information coding by populations of neurons as well as interactions between populations, both within as well as across areas, are related is very noteworthy.

As far as I can judge, in general the research presented here is sound. The last author is an eminent figure in the field of two-photon imaging and parts of the data have been recently published in a high-impact, peer-reviewed journal (Han and Helmchen, Nat Neurosci, 2024). Data analysis mostly appears to be sound, with methods used being well established in the field and applied appropriately.

Response:

Thank you for evaluating our manuscript and for the positive comment.

That said, I have a few comments/questions/concerns regarding aspects of the data analysis:

1. Mice performed different numbers of trials each day. As the authors write, “To account for the different trial numbers each day, we split each training day into sessions of 80-120 trials.” I’m wondering how this procedure affects the analysis outcome, since in this way data from the same training session will be mixed with data from another session. Thus, data samples are not equally independent which will affect the variance of the data. Why didn’t the authors simply stratify trial numbers across training days, sampling from the daily outcome of 100-500 trials that they had (taking e.g. n=100 trials per day)? My best guess is that this won’t badly affect the most robust findings of their study. I hence would like to ask for a re-analysis of the data after subsampling and stratification across training days.

Response:

Thank you for this suggestion. We performed the suggested analysis by taking 120 trials from each training day, therefore avoiding correlated sessions. More specifically, as the behavioral performance fluctuated within each day, we split each training day into consecutive sessions of 120 trials and took the session with the highest performance within that day. Using these stratified data, we reproduced the main findings. The results are shown in a new supplemental figure.

Specific changes in the revised manuscript:

- We added a new Supplemental Fig. 11 with analysis results using one session per day with matching trial numbers, and included the results in the manuscript:

“As this procedure pooled correlated neuronal populations from the same training day as independent samples, we also stratified one session per mouse with the same number of trials for each day, which reproduced our main findings (Supplemental Fig. 11).” (Line 577-579)

2. In Figure 2, changes in the percentages of task-responsive and task-discriminative neurons are compared. It is not mentioned anywhere that one can expect a certain number (percentage) of neurons to be responsive to task variables (and hence to differentiate between them) or not just by chance. Please make that clear in the text and indicate the related significance threshold assumed (5%, say) in the figures.

Response:

Thank you for this comment. In our analysis, the identified task-responsive and task-discriminative neurons were non-random; all identified neurons passed statistical testing procedure, and the chance level

Extra Figure 4. Percentage of responsive and discriminative neurons in shuffled data. Left: percentage of responsive neurons in shuffled data; Right: percentage of discriminative neurons in shuffled data. No neurons were found in either cases.

is zero. In more detail, neurons were defined as task-responsive if their activity level is significantly higher in a specific task window compared to a null distribution generated by randomly sampling frames outside of the task window, with matching numbers of frames (100 shuffles). Within this pool of responsive neurons, discriminative neurons were defined as neurons that showed significantly higher activity in one task condition (e.g., texture 1 vs. 2). This procedure was performed for each neuron in the population independently, and the null distribution was generated from the out-of-window activity within the same neuron. When we shuffled the activity of each neuron within sessions and performed the same procedure, we found zero responsive and discriminative neurons for all task phases and all populations (see Extra Figure 4 below). Therefore, the reported percentage of task neurons represent non-random (above chance level) findings.

Specific changes in the revised manuscript:

- We added a corresponding explanation in the methods section to clarify this point:

“Responsive and discriminative neurons identified with this method are non-random, as no significant neurons were identified in shuffled data, where the activity of individual neurons was randomized within sessions.” (Line 660-661)

3. What was the reason to select precisely 30 PCs of the subpopulation dimensionality before subspace computation?

Response:

The main reason we performed PCA on the population activity before subsequent analysis was to ensure stable CCA model estimation. CCA requires a sufficient number of samples to generate stable solutions. Our dataset consisted of ~120 trials per session, and ~10 time points for each task window. A previous study using simulated datasets has shown that ~50 samples per variable are required to generate a stable solution (Helmer et al, 2024). This results in roughly 30 variables per area to ensure stable model estimation. To verify the reproducibility of our results with different PC numbers, we re-analyzed the data

using top 60 PCs, which retained more variance. We were able to reproduce the main findings using 60 PCs. The results are shown in a new supplemental figure.

Specific changes in the revised manuscript:

- We added a new Supplemental Fig. 4 with the main results using top 60 PCs, and included a corresponding text:

“Using more PCs did not change the main findings in the following sections (Supplemental Fig. 4).” (Line 204-205)

- We revised the results and methods sections to explain our considerations of choosing 30 PCs:

“To reduce the computational complexity and ensure stable model estimation^{23,36}, we reduced the subpopulation dimensionality with PCA and kept the top 30 PCs prior to the subspace computation.” (Line 202-204)

“Then, to reduce the computational challenge and to ensure a stable CCA computation, we reduced the dimensionality of each subpopulation with PCA. CCA requires a sufficient amount of samples to generate stable solutions. Our dataset consisted of ~120 trials per session, and ~10 time points for each task window. A previous study using simulated datasets has shown that ~50 samples per variable are required to generate a stable solution³⁶. To ensure such a condition is met, and to ensure that each subpopulation had a similar number of variables, especially for the inter-area axis computation, we kept the top 30 PCs for each area.” (Line 670-676)

- We also added the retained variance in the methods section:

“The top 30 PCs explained total variance of 41.4 ± 2.0 (naive, mean \pm SEM, in %), 41.7 ± 1.5 (learning), 51.7 ± 2.2 (expert, same order for the following) in SI, 37.0 ± 2.6 , 38.5 ± 2.6 , 47.1 ± 2.5 in PPC-RL, and 38.0 ± 1.6 , 42.7 ± 1.4 , 50.8 ± 2.9 in PPC-A. Taking top 60 PCs did not change our main results (Supplemental Fig. 4; explained variance: 63.6 ± 2.0 , 65.6 ± 1.7 , 75.1 ± 2.1 in SI, 58.5 ± 2.9 , 60.3 ± 2.7 , 72.0 ± 2.5 in PPC-RL, 62.2 ± 2.2 , 68.3 ± 1.7 , 77.6 ± 2.8 in PPC-A).” (Line 676-680)

4. In Figures 5-7, image plots of data matrices are shown including statistics for tests of “real” data against a shuffle distribution (red boxes) as well as asterisks indicating a significantly higher value in comparison with the respective contrasted condition (e.g., naïve vs. expert). In some of these matrix fields, there are asterisks indicating a significantly higher or lower value while at the same time, neither is surrounded by a red box. Does that mean that neither condition has a value higher or lower than the shuffle distribution but when compared against each other, one is significantly higher than the other? But then, what is being compared here; merely noise? Please clarify this and only consider as valid those comparisons where at least one side of a pair is significant as compared to the shuffle.

Response:

Thank you for this comment. We agree that the comparisons can only be meaningfully interpreted when at least one side of the pair is significant as compared to the shuffle. We have revised all relevant panels to restrain the comparison to significant entries.

Specific changes in the revised manuscript:

- We revised Fig. 5f,h, Fig. 6e-i, Fig. 7f-h, and only considered comparisons where one at least one side of a pair was significant as compared to the shuffle. This was also applied to Supplemental Fig. 4, 5, 7, 11.

- We added the corresponding information in the methods section:

“Subsequent statistical comparisons between naive and expert conditions were restrained to pairs where at least one side was significantly above shuffled distribution.” (Line 721-723)

The above comments/concerns/requests notwithstanding, the present work mostly supports the conclusions provided and the claims raised, requiring no further basic evidence. However, I would like the discussion to be more focused on the truly novel aspects of the results and how the study expands our current understanding of how neuronal population dynamics shape learning processes.

Response:

Thank you for this comment. Our dataset provides one of the first biological applications of state-of-the-art imaging techniques, which allowed us to study learning-related changes on multiple levels, from single-neuron level, to local populations and cross-area interactions. As one of the first in vivo and longitudinal studies that covered all these scales and behavioral complexity, we provided a panoramic characterization of learning-related changes on these different levels. In addition, we simultaneously observed a primary sensory area S1 and two subregions of a higher association area PPC, and we found distinct changes in these areas during learning, supporting the idea of differential learning mechanisms across brain regions, which also depends on the specific task studied. We have revised our discussions and emphasized these novel aspects.

Specific changes in the revised manuscript:

- We revised the discussion to reflect the novel aspects of the results:

“Using simultaneous multi-area two-photon imaging, combined with a sensory discrimination task and longitudinal behavioral and neuronal recordings throughout learning, we provided a systematic overview of learning-related changes in S1 and the higher association areas PPC-A and PPC-RL, both required for optimally performing the task. We discovered changes on different levels, supporting multiple learning mechanisms.” (Line 334-338)

“In our study, we found evidence for both mechanisms. As the task representation improved, the within-area subspaces in PPC also strengthened and expanded, and their alignment with the task encoding dimensions increased. We also observed distinct changes in S1, PPC-RL, and PPC-A that accompanied the improved task representation during learning. S1 and PPC-RL showed decreased neuronal activity dimensionality, decreased within-area axis similarity across time, but stable trial-to-trial variability and increased encoding axis similarity. These findings potentially suggest that the populations compress their variability at each time point to achieve more consistent task encoding, with a dynamic shared variability structure across time. In contrast, PPC-A showed decreased trial-to-trial variability, decreased overall encoding axis similarity, but stable neuronal activity dimensionality and stable within-area axis similarity across time. These adaptations in PPC-A potentially suggest reduced noise levels across trials and diversification of the task-related encoding structure across time, while preserving the variability of the population.” (Line 348-359)

“While both PPC-RL and PPC-A were engaged in this task and required for optimal task performance, learning this task preferentially involved PPC-A throughout the tone, texture and choice windows, opening additional within-area interaction subspaces, whereas PPC-RL was involved mostly in texture processing. In addition, the interaction subspaces between S1 and PPC-A expanded throughout the task, whereas S1 and PPC-RL interaction was enhanced only during the texture window. These results agree with our previous findings that PPC-A, but not PPC-RL, encodes predictive texture information using the preceding tone, in expert mice performing the same behavioral task²³.” (Line 372-378)

Taken together, I in principle endorse the publication of the present work in Nature Communications and thank the authors for the effort they put into it. Please adequately address the questions/concerns raised above beforehand.

Thank you.

Response to the Reviewers' comments, Han and Helmchen, manuscript NCOMMS-24-59079-A

Dear Reviewers,

Thank you for your review of our manuscript “Coordinated multi-level adaptations across neocortical areas during task learning”. We appreciate your effort in reviewing our manuscript. We have revised our manuscript to address your comments and concerns. In addition, we made minor edits to the text, and corrected the animal and session numbers in a few places in the text and figure legends. By taking into account your inputs, we believe the manuscript is now more rigorous and more clearly presented.

In the following, we list the original comments from the reviewers, followed by our responses (in blue), and how we have addressed the points in the revised manuscript (in orange). We appreciate your evaluation of our responses and the revised manuscript. Besides this letter and a clean version of the revised manuscript, we also attach a version with tracked changes (major changes highlighted in red) for your convenience.

Thank you again for your time and effort.

Zurich, 5th May 2025

Shuting Han

Fritjof Helmchen

REVIEWER COMMENTS

Reviewer #1 (Remarks to the Author):

I appreciate the authors' revisions in response to my comments as well as those from other reviewers.

1) However, I find the authors' claim that anticipatory whisking or facial movement is inherently part of task learning to be somewhat rough and overly broad, especially considering that "task learning" in this study appears to focus on cognitive task learning—particularly processes related to "sensory processing and decision-making" as stated in the abstract.

The authors state: "We still observed increased facial movement during the texture window even without early licking frames; however, this is likely due to the enhanced and anticipatory whisking that emerges during learning ... This change in whisking pattern is part of task learning and thus is not in conflict with our neural findings but rather could serve as one mechanism that explains certain neural findings."

This response conflates general behavioral adaptation with task-specific cognitive learning while failing to carefully distinguish between them. If any behavioral change occurring during training is categorized as "task learning," it raises the question of why the task was designed to require specific cognitive components in the first place. Under such a broad definition, there is a risk that some of the observed neural changes may reflect epiphenomena related to motor adaptation rather than the cognitive processes that the task was intended to assess.

To be clear, I am not suggesting that all observed neural changes and inter-regional dynamics can be attributed to movement-related factors. However, the authors should at least explicitly acknowledge that motor coordination, alongside task-related sensory processing and decision-making, could influence the neural findings. Clarifying this distinction would strengthen the interpretation of their results.

Response:

Thank you for this thoughtful and important comment. We agree that it is crucial to distinguish between motor adaptation and cognitive aspects of task learning, particularly in interpreting neural changes. In our task, successful performance typically involves active whisking for texture sensation (see Chen et al., 2015; Gilad et al., 2018), and thus learning inherently involves both motor and cognitive components. We acknowledge that some of the observed neural dynamics may reflect motor-related changes, and we have revised the manuscript to make this clearer.

However, we do not believe that all neural adaptations are attributable to motor learning. Several of our findings, such as the recruitment of task neurons, reduced response variability, enhanced encoding of task variables, and improved information flow, agree with observations from learning paradigms that do not involve movement related to self-initiated sensation, for example, with visual stimuli or closed-loop optogenetics stimulation (Poort et al., 2015; Failor et al., 2022; Haimerl et al., 2023; Yan et al., 2014; Pancholi et al., 2023). These parallels suggest that at least part of the neural changes observed here reflect cognitive aspects of task learning rather than motor adaptation alone. We have updated the discussion to clarify these points and to indicate the potential influence of motor coordination on our results.

Specific changes in the revised manuscript:

- We have modified the discussion part to reflect the potential contribution of movement adaptations:

“We would like to note here that learning a tactile discrimination task is typically accompanied by refined movement patterns^{2,3,5,6,30}, which could influence the observed neuronal activity. In our case, the neuronal findings were accompanied by changes in facial movement during the tone and texture windows, possibly due to specific whisking patterns that emerged during learning^{6,30}. These changes in movement patterns could potentially explain some of the neuronal findings, alongside with the neuronal adaptations underlying sensory processing and decision making related to the task.”
(Line 382-387)

2) I also appreciate the authors sharing the optogenetic silencing results of mismatched trials in expert mice as Extra Fig. 1. I agree that, while informative, this analysis is not essential to the current study’s focus on learning-related changes, and its inclusion is not necessary.

Response:

Thank you for this assessment.

3) Although the authors mentioned that they have redesigned Figures 5–7, many subpanels are still too small, significantly reducing clarity. If the goal is to effectively present these data to readers, space should not be a limiting factor.

Response:

Thank you for this comment. We have redesigned Figures 5-7, and we hope that the subpanels are sufficiently large now.

Specific changes in the revised manuscript:

- We have updated Fig. 5-7.

Reviewer #3 (Remarks to the Author):

I appreciate the effort the authors put into the revised version of the manuscript. From my side, almost all the issues I've raised have been addressed to my satisfaction.

There is one aspect though that escaped my attention during the first round and that I now noticed. In Figure 2 panels d and e (lines 147-156), the authors report the percentages of neurons being responsive to task variables and task-discriminative. They then go on to make statements about percentages of responsive and discriminative neurons changing over the course of learning. They provide non-parametric test statistics to underscore their claims, but I don't think that this is the right statistic here. As far as I can see, a Chi-square or similar test statistic would be needed to compare percentages of responsive or discriminative neurons here. Please comment on this and provide the appropriate analyses and results to back up the original claims.

Response:

Thank you for this careful observation. In Figure 2d and 2e, our objective was to evaluate whether the proportion of task-responsive and task-discriminative neurons changed across learning stages, in separate task windows including tone, texture, choice and reward windows. To do this, we calculated, for each

session, the percentage of neurons falling into these categories and then compared these percentages between Naive, Learning, and Expert stages. Since the number of sessions differed across learning stages, and the data were not normally distributed, we chose an unpaired non-parametric test (Wilcoxon rank-sum test) to assess statistical significance. P-values were computed for Naive versus Learning sessions, and for Naive versus Expert sessions. Comparisons were made separately for neuron percentage from each task window (tone, texture, choice, reward), and were corrected for multiple comparisons using false discovery rate. This approach treats each session as an independent data point and allows us to assess whether task neuron percentage significantly changed across learning stages, despite variability in the number of recorded neurons per session.

As far as we understand, a Chi-square test of independence is used to determine the statistical difference between two categorical variables. A Chi-square test would be appropriate only if all neurons were pooled across sessions, after individually classified into discrete task categories within each session. However, our analysis aggregates neuronal data at the session level and treats the session as the unit of comparison (rather than individual neurons pooled across sessions). Consequently, we analyzed continuous variables (session-level percentages of task-tuned neurons), making non-parametric testing more appropriate than a Chi-square approach. We believe that Chi-square test is not well suited for our analysis.

To further address the reviewer’s concern, we performed Chi-square test on the pooled neurons across sessions anyways. We aggregated neurons from all sessions and labeled each neuron with a task window category (tone, texture, choice, reward, or none), as well as a learning stage (naive, learning, expert). We treated each neuron as independent samples (note that this assumption is not true, since some neurons were recorded across multiple days). We observed very similar results as shown in Figure 2d and 2e and

Extra Figure 1. Chi-square statistics on task neurons. (a) Percentage of task-responsive neurons from pooled neuron across all sessions. Note no error bar is available. Chi-square test of independence was performed. (b) Percentage of task-discriminative neurons from pooled neuron across all sessions. (***) $p < 0.001$; Bonferroni corrected).

stronger statistical significance (see Extra Figure 1a, b). We hope this additional analysis resolves the reviewer's concern.

As we believe that our original statistical test is suitable for our purpose, we did not include this figure in the revised manuscript. We can include it if the reviewer feels that it would be appropriate and helpful to strengthen the statistical significance of the results.

Specific changes in the revised manuscript:

- We revised the legend of Figure 2 to include more details:

“Statistical tests in d, e, g were performed using Wilcoxon rank-sum test between Naive and Learning sessions, and between Naive and Expert sessions, using pooled session data”

Thanks much again for your efforts!

Thank you.